# Natural variation of *Dt2* determines branching in soybean

Qianjin Liang [1,2], Liyu Chen[3], Xia Yang[1,2], Hui Yang[3], Shulin Liu [1], Kun Kou[3], Lei Fan[1], Zhifang Zhang[1], Zongbiao Duan[1], Yaqin Yuan[1,2], Shan Liang[1,2], Yucheng Liu[1], Xingtong Lu[1,2], Guoan Zhou[1], Min Zhang[1], Fanjiang Kong [3] ✉ & Zhixi Tian [1,2] ✉

Shoot branching is fundamentally important in determining soybean yield. Here, through genome-wide association study, we identify one predominant association locus on chromosome 18 that confers soybean branch number in the natural population. Further analyses determine that *Dt2* is the corresponding gene and the natural variations in *Dt2* result in significant differential transcriptional levels between the two major haplotypes. Functional characterization reveals that Dt2 interacts with GmAgl22 and GmSoc1a to physically bind to the promoters of *GmAp1a* and *GmAp1d* and to activate their transcription. Population genetic investigation show that the genetic differentiation of *Dt2* display significant geographic structure. Our study provides a predominant gene for soybean branch number and may facilitate the breeding of high-yield soybean varieties.

Shoot branching is fundamentally important to the growth and productivity of crops[1]. By unconscious or conscious selection of the superior alleles of key genes that confer branching architecture, humans have significantly increased crop yields. One typical example is the selection and utilization of the *TEOSINTE BRANCHED1* (*TB1*) gene during the domestication of maize[2]. *TB1* and its orthologs play profound roles in modulating branching architecture via determination of the bud activation potential in plants[1,3–5]. Cultivated maize (*Zea mays*) was domesticated from teosinte (*Z. mays* ssp. *parviglumis*). Largely benefitting from the selection of a higher-expression allele of *TB1*, cultivated maize was successfully domesticated as a high-yield crop with a single culm, in contrast to its highly branched ancestor teosinte[6]. Another promising advance is the application of *IPA1* in rice. A point mutation in *IPA1* leads to an ideal rice plant with fewer tillers, increased plant height, lodging resistance, and thus enhanced grain yield[7–9]. By introducing beneficial *ipa1* alleles into widely cultivated cultivars, a series of new elite varieties with higher yields were developed[7]. An increasing number of regulatory genes that control shoot branching are being identified[1,10], providing genetic resources for breeding high-yield crops via molecular design.

Soybean (*Glycine max* (L.) Merr.) is one of the most important crops that supplies more than half of global oilseed production and approximately one quarter of the world's plant protein[11]. With the increasing population and continuous improvement in people's living standards, it was estimated that the soybean yield has to be doubled by 2050 to meet the consumption demands[12]. In contrast to the dramatic increase in yield of major crops, such as rice and wheat, which greatly benefitted from the Green Revolution, soybean yield has not been improved significantly in the past six decades. Modulating branch number, one of the most profound traits that determines the final yield of soybean, is crucial for high-yield soybean breeding[13]. However, to date, the genes associated with soybean branching have seldom been reported[14].

Here, we show that natural variation in *Dt2* predominantly determine soybean branching. We also reveal that Dt2 interact with GmAgl22 and GmSoc1a to bind to the promoters of *GmAp1a* and *GmAp1d* to regulate their transcriptions. Interestingly, the selection of

[1]State Key Laboratory of Plant Cell and Chromosome Engineering, Institute of Genetics and Developmental Biology, Chinese Academy of Sciences, Beijing, China. [2]University of Chinese Academy of Sciences, Beijing, China. [3]Guangdong Provincial Key Laboratory of Plant Adaptation and Molecular Design, Innovative Center of Molecular Genetics and Evolution, School of Life Sciences, Guangzhou University, Guangzhou, China. ✉e-mail: kongfj@gzhu.edu.cn; zxtian@genetics.ac.cn

*Dt2* is associated with geographic differentiation. Modulating *Dt2* lines resulted in significantly increase of soybean adaptation and yield, which is also associated with its effect on maturity.

## Results

### Natural variation in *Dt2* predominantly determines soybean branching

To identify the key genes that control branch number in soybean, we phenotyped 2409 accessions from our previous resequencing panel[15] in 2017 and 2018. Phylogenetic and principal-component analysis of the landraces and cultivars did not show significant genetic differentiation (Supplementary Fig. 1a, b). Genome-wide association study (GWAS) performed using a mixed linear model revealed a stable association signal across the 2 years in a 40 kb interval block on chromosome 18 (Fig. 1a–d and Supplementary Fig. 1d–g). Within this 40 kb interval, a total of 5 protein-coding genes were annotated according to the reference genome ZH13[16,17] (Fig. 1d), among which *SoyZH13_18g242900* showed higher specific expression at the shoot apical meristem (Supplementary Fig. 2a), a tissue closely related to the final branching architecture[1]. Therefore, *SoyZH13_18g242900* was

considered to be the candidate gene controlling branch number in soybean.

*SoyZH13_18g242900*, also known as *Dt2*, has been found to play important roles in regulating multiple agronomic traits, including stem growth habit, plant height, and flowering time[18–20]. Whether it could modulate branch number has not been investigated. Phylogenetic analysis found that *Dt2* belongs to MADS-box transcription factor family and shares high homology with *AGL79* in *Arabidopsis*, a member of the AP1/FUL subfamily (Supplementary Fig. 2c). Of the association polymorphisms with minor allele frequency (MAF) >0.05, two SNPs from the promoter regions (3259 bp and 2580 bp upstream of the translation start site, respectively) and six SNPs from the introns showed higher association values than the threshold, and a G/A SNP from the first exon that changed amino acid serine to asparagine showed a lower association value than the threshold (Supplementary Fig. 2b). Based on these higher-value association SNPs and the G/A nonsynonymous SNP, three major haplotypes of *Dt2*: *Dt2*[HapI-1], *Dt2*[HapI-2], and *Dt2*[HapII] were classified in the natural population (Fig. 1e, f). We found that the accessions harboring *Dt2*[HapI-1] and *Dt2*[HapI-2] did not exhibit a significant difference in branch number, whereas *Dt2*[HapII] showed a significantly lower branch number than *Dt2*[HapI-1] and *Dt2*[HapI-2] (Fig. 1f).

Transient transcription activity assays suggested that the promoter sequence of *Dt2*[HapII] had significantly higher transcriptional activity than that of *Dt2*[HapI] (Fig. 1g). We then randomly selected 20 representative natural accessions and investigated the gene expression level of *Dt2* and branch number. A negative correlation between the *Dt2* expression level and branch number was observed (Supplementary Fig. 3a). Investigation of a pair of *Dt2* near-isogenic lines (NILs) also revealed that the *Dt2*[HapII] line exhibited a significantly decreased branch number than the *Dt2*[HapI-1] line (Supplementary Fig. 3b, c). Consistently, quantitative real-time PCR (RT-qPCR) assays showed that *Dt2* expression in the *Dt2*[HapII] line was significantly higher than that in *Dt2*[HapI-1] (Supplementary Fig. 3d). The above results suggested that the variation at the promoter of *Dt2* played an important role in determining branch number in soybean.

### Functional validation of the role of *Dt2* in controlling branch number

To validate the function of *Dt2* in determining branch number, we knocked out *Dt2* in Dong Nong 50 (DN50), a variety harboring *Dt2*[HapII] with a mean of four branches, by the CRISPR/Cas9 system and obtained two independent homozygous knockout lines (named *Dt2*[CR-1] and *Dt2*[CR-2], respectively) (Supplementary Fig. 4). Field characterization demonstrated that the *Dt2*[CR] lines exhibited increased branch number compared with the wild type DN50 (Fig. 2a, b, f). The *Dt2*[CR] lines also showed significantly delayed flowering and maturity, and increased plant height and stem node number, which were consistent with its reported functions[18,19] (Fig. 2g and Supplementary Fig. 5a–c). Moreover, we found that the *Dt2*[CR] lines exhibited multiple yield related trait changes, including higher 100 seed weight, longer seed length and width, and higher grain weight per plant, resulting in significantly increased yield per plot (Fig. 2h and Supplementary Fig. 6a, c–f).

We further overexpressed the coding sequence (CDS) of *Dt2* driven by the *35S* promoter in DN50 and obtained two independent transgenic overexpression lines (named *Dt2*[OE-1] and *Dt2*[OE-2], respectively). RT-qPCR assays showed that the expression of *Dt2* was increased approximately threefold in the *Dt2*[OE] lines compared to the wild type DN50 (Fig. 2e). In contrast to the results from the *Dt2*[CR] lines, the *Dt2*[OE] lines exhibited decreased branch number, promoted flowering time (Fig. 2c, d, f, h) and maturity, decreased plant height, decreased stem node number (Supplementary Fig. 5), decreased pod number per plant, and decreased seed length and width, thus exhibiting decreased yield per plot (Fig. 2h and Supplementary Fig. 6b, c–f). These experiments showed that *Dt2* negatively regulated branch number in soybean. Growth state statistics of axillary buds in leaf axils

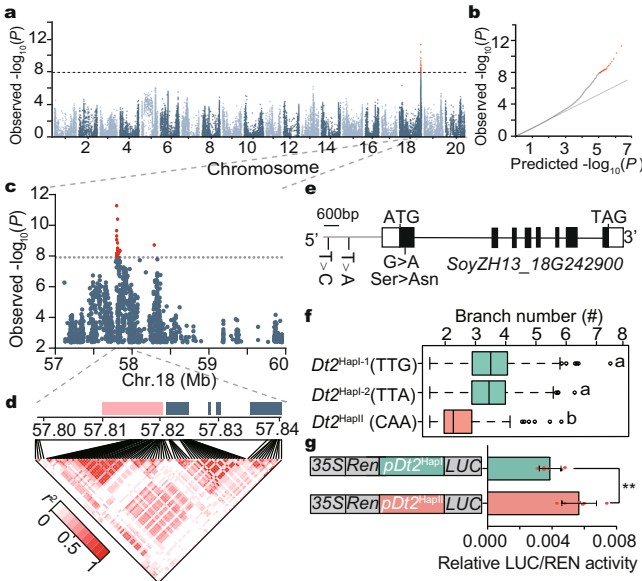

**Fig. 1 | GWAS of branch number and haplotype analysis of Dt2 in soybean. a** Manhattan plot of GWAS for branch number using 2 years of Blup data of the 2409 accessions. *P* values are calculated based on linear mixed model in GWAS and the dashed horizontal line indicates the genome-wide significance threshold ($P = 1 \times 10^{-7.9}$), which is determined by the Bonferroni test. $-\log_{10} P$ values are plotted against the position of SNPs on 20 chromosomes. **b** Quantile-quantile plot for the branch number. For quantile-quantile plot, $-\log_{10}$-transformed observed *P* values are plotted against $-\log_{10}$-transformed expected *P* values. **c** Genome-wide Manhattan plot in the 57–60 Mb region on chromosome 18. The red lead SNPs are shown above the threshold signals. **d** Linkage disequilibrium plot for SNPs in the 57.80–57.84 Mb region from a continuous association block. Navy blue box, genes. Pink box, candidate gene *Dt2*. Asterisk, position of the peak SNP. The color key (white to red) represents linkage disequilibrium value ($r^2$) accessions. **e** Gene structure of *Dt2*. Two SNPs (−3259th T > C, −2580th T > A) in the promoter and a nonsynonymous SNP (+98th G > A) in the first exon are labeled on the gene sketch. **f** Haplotype analysis of *Dt2*[HapI-1] (*n* = 1052 accessions), *Dt2*[HapI-2] (*n* = 262 accessions) and *Dt2*[HapII] (*n* = 179 accessions). In each box plot (drawn by R 4.1.1 software), the center line indicates the median, the edges of the box represent the first and third quartiles, and the whiskers extend to span a 1.5 interquartile range from the edges. Different letters indicate statistically significant differences at *P* < 0.05 by one-way ANOVA test. **g** Promoter activity analysis of *Dt2*[HapI] and *Dt2*[HapII] using sequences 3,400-bp upstream from the translation initiation site (*n* = 5 biologically independent replicates), **P < 0.01, two-sided *t*-test, and $P = 6.0 \times 10^{-3}$. Data in (**f**, **g**) are the mean ± SEM. Source data are provided as a Source Data file.

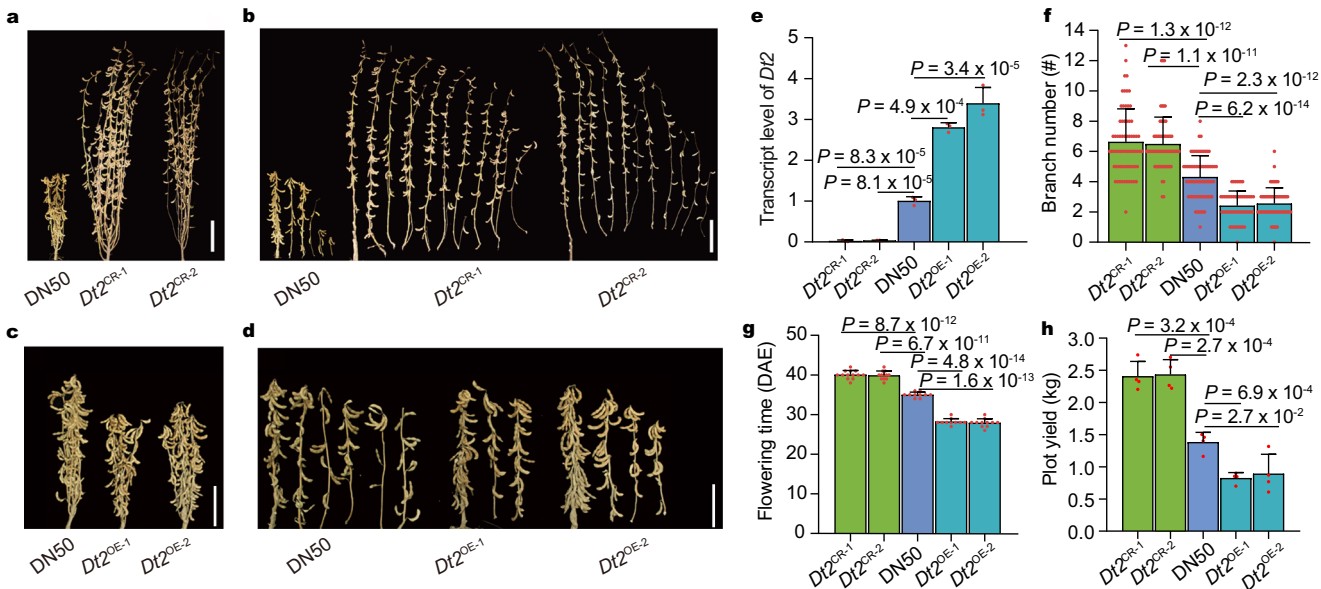

**Fig. 2 | Dt2 negatively regulates the branch number. a** Architecture of DN50 and *Dt2*^CR plants at the harvest stage. **b** Branching ability of DN50 and *Dt2*^CR after harvest. **c** Architecture of DN50 and *Dt2*^OE plants at the harvest stage. **d** Branching ability of DN50 and *Dt2*^OE after harvest. **e** Relative expression level of *Dt2* in DN50 and different transgenic lines (*n* = 3 biologically independent replicates). **f** Branch number of DN50, *Dt2*^CR and *Dt2*^OE lines (*n* = 70 biologically independent plants).

**g** Flowering time of DN50, *Dt2*^CR and *Dt2*^OE lines (*n* = 30 biologically independent plants). DAE indicates days after emergence. **h** Plot yield of DN50, *Dt2*^CR and *Dt2*^OE (*n* = 4 biologically independent replicates). The plot area is 10 m². Scale bars in the (**a–d**) are 10 cm. Data in (**e–h**) are the mean ± SEM. Statistical significance is determined using a two-sided *t*-test. Source data are provided as a Source Data file.

between DN50 and *Dt2*^CR lines showed that the effect of *Dt2* on branch development may be related to inflorescence determinacy (Supplementary Fig. 7).

### Dt2 interacts with GmAgl22 and GmSoc1a

Transcriptional profiling of different tissues showed that *Dt2* was highly expressed in the lateral buds and shoot tip (Supplementary Fig. 8a). An in situ hybridization assay demonstrated that *Dt2* was specifically expressed in the axillary meristem and shoot apical meristem (Supplementary Fig. 8b). To identify the protein interaction partners of Dt2, we carried out a yeast two-hybrid (Y2H) assay by screening the cDNA library constructed from the lateral buds and identified 136 unique cDNA clones that corresponded to 28 genes (Supplementary Table 1).

Of the interacting proteins, an ortholog of *Arabidopsis* SHORT VEGETATIVE PHASE (SVP)/AGL22, named *GmAgl22* in this study, showed more repeats and was of interest to us. *GmAgl22* encodes a MADS-box transcription factor in *Arabidopsis* and functions as a floral repressor in the thermosensory pathway[21–23]. The interaction between Dt2 and GmAgl22 was further confirmed by point-to-point Y2H, bimolecular florescence complementation (BiFC), split luciferase complementation (split-LUC) and coimmunoprecipitation (Co-IP) assays (Fig. 3a–c and Supplementary Fig. 9a, b). Detailed investigation demonstrated that the K-box domain of Dt2 was essential for the physical interaction between Dt2 and GmAgl22 (Supplementary Fig. 9d). In addition, we found that GmAgl22 could interact with itself in Y2H, split-LUC and Co-IP assays (Supplementary Fig. 10a–c), indicating that the protein develops dimers or polymers to exert its function. A previous study reported that Dt2 could interact with GmSoc1a to affect the soybean growth habit[18,24]. We suspected that GmAgl22 could also interact with GmSoc1a, which was then confirmed by Y2H, BiFC, split-LUC and Co-IP assays (Fig. 3d–f and Supplementary Fig. 9c). Expression pattern analysis found that *GmAgl22* and *GmSoc1a* were highly expressed in the lateral bud and shoot tips (Supplementary Fig. 10d, e). In situ hybridization assays demonstrated that *GmAgl22* and *GmSoc1a*, similar to *Dt2*, were specifically expressed in

the axillary meristem in different developmental stages (Supplementary Fig. 11).

Previously, we generated a *GmSoc1a* knockout mutant by the CRISPR/Cas9 system (named *GmSoc1a*^CR) and determined that the mutation could significantly affect flowering time. We also obtained two *GmAgl22* overexpression lines (named *GmAgl22*^OE-1 and *GmAgl22*^OE-2) (Supplementary Fig. 10f). We then compared the branch numbers of *GmSoc1a*^CR and *GmAgl22*^OE with those of non-transgenic parents, respectively. We found that the *GmSoc1a*^CR line showed significantly increased branch number (Fig. 3g, h), and the *GmAgl22*^OE lines showed significantly decreased branch number (Fig. 3i, j). These results indicated that Dt2, GmAgl22 and GmSoc1a may function by forming a complex to control branching in soybean.

### Dt2 regulates the expression of the GmAp1 gene family

To mine the downstream targets and regulatory network of *Dt2*, transcriptome profiling by RNA-Seq was performed with the lateral buds from WT (DN50), *Dt2*^CR and *Dt2*^OE. Using a *P* value < 0.05 and the fold change larger than 2 as thresholds, 646 up-regulated and 296 down-regulated genes were identified as differentially expressed genes (DEGs) between the wild type and *Dt2*^CR lines (WT/*Dt2*^CR) and 1160 up-regulated and 464 down-regulated genes were identified between the wild type and *Dt2*^OE lines (WT/*Dt2*^OE) (Supplementary Fig. 12a–d). Gene Ontology (GO) term analysis demonstrated that the DEGs from the WT/*Dt2*^CR and WT/Dt2^OE lines were enriched among metabolic process terms. Notably, multiple terms were related to carbohydrate metabolism and photosynthesis processes (Supplementary Fig. 12e, f), which was consistent with previous findings that sugar metabolism or signaling played an important role in axillary bud outgrowth[1,25,26].

We then further narrowed down the DEGs by selecting the genes with an opposite pattern in the WT/*Dt2*^OE and WT/*Dt2*^CR panels: (1) up-regulated in the WT/*Dt2*^OE panel but down-regulated in the WT/*Dt2*^CR panel, and (2) down-regulated in the WT/*Dt2*^OE panel but up-regulated in the WT/*Dt2*^CR panel. In total, 30 genes meeting these criteria were identified (Fig. 4a). Interestingly, half of these 30 genes were annotated as agamous-liked genes, and most of these agamous-liked genes

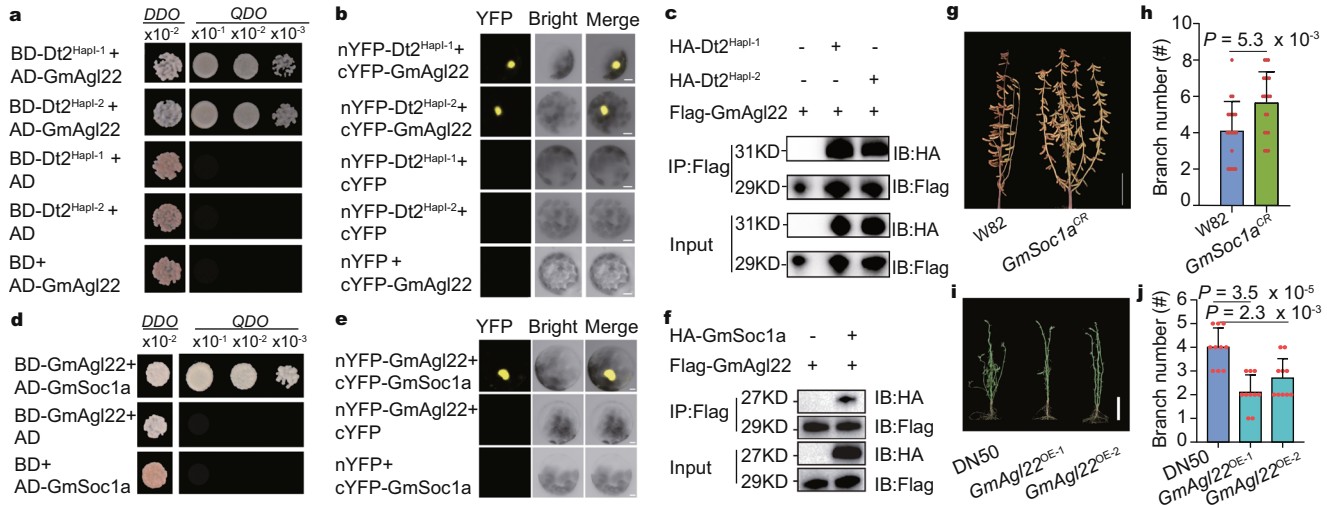

**Fig. 3 | Dt2, GmAgl22 and GmSoc1a could interact with each other and affect the branch number. a** Interaction between Dt2 and GmAgl22 in the yeast two-hybrid assay. Transformed yeast cells are grown on *DDO* (Trp/Leu) or *QDO* (Trp/Leu/His/Ade) synthetic dropout medium. The number at the top indicates three serial dilutions. AD, GAL4 activation domain; BD, GAL4 DNA-binding domain. **b** BiFC analysis of the interaction between Dt2 and GmAgl22. Scale bars, 10 mm. cYFP C-terminal portion of YFP. nYFP N-terminal portion of YFP. **c** Co-IP analysis of the protein interactions between Dt2 and GmAgl22. Flag-tagged GmAgl22 is co-transformed with HA-tagged Dt2 into *Arabidopsis* protoplasts. **d** Interaction between GmSoc1a and GmAgl22 in the yeast two-hybrid assay. **e** BiFC analysis of the interaction between GmSoc1a and GmAgl22. Scale bars, 10 mm. **f** Co-IP analysis of the protein interactions between GmSoc1a and GmAgl22. **g** Phenotypic comparison between W82 and *GmSoc1a*^CR lines. Scale bars, 20 cm. **h** Branch number statistic in W82 and *GmSoc1a*^CR lines (*n* = 20 biologically independent plants). **i** Phenotypic comparison between DN50 and *GmAgl22*^OE lines. Scale bar, 10 cm. **j** Branch number statistic in DN50 and *GmAgl22*^OE lines (*n* = 10 biologically independent plants). All the data in the graphs represent the mean ± SEM. Statistical significance is determined using a two-sided *t*-test. For (**b**, **e**), at least 5 independent cells are observed and a representative result is shown. For (**c**, **f**), at least 3 independent replicates are performed and a representative result is shown. Source data are provided as a Source Data file.

showed higher expression in the shoot meristem and flower (Supplementary Fig. 13a). Further comparison of the DEGs with previously reported *Dt2* ChIP-seq data[20] revealed that 4 genes might be the direct targets of *Dt2*: *SoyZH13_16G083100* (*GmAp1a*), *SoyZH13_01G060300* (*GmAp1c*), *SoyZH13_02G112700* (*GmAp1d*) and *SoyZH13_18G251400* (*GmRPP13*). The three *GmAp1* gene family members were significantly up-regulated in the *Dt2*^OE lines and down-regulated in the *Dt2*^CR line, whereas *GmPRR13* was down-regulated in the *Dt2*^OE lines and up-regulated in the *Dt2*^CR lines (Fig. 4a and Supplementary Fig. 13b). In this study, we focused on a functional assay of *GmAp1* gene family using *GmAp1a* and *GmAp1d* as representatives. A previous in situ hybridization assay demonstrated that *GmAp1a* was specifically expressed in the shoot apices in the V2 stage[27], and we also found that *GmAp1a* had a similar expression pattern as *GmAp1d* (Supplementary Fig. 13c), indicating that they may perform a similar function in branching development.

A previous study showed that Dt2, a MADS-box protein, could bind to CArG elements[20]. Putative binding motif prediction using PlantPAN 3.0 (http://plantpan.itps.ncku.edu.tw/) indicated that a sequence located −1274 bp from the *GmAp1a* translation initiation site (named Probe1) (Fig. 4b) and a sequence located −678 bp from the *GmAp1d* translation initiation site (named Probe2) (Supplementary Fig. 14a) might be the target sites of Dt2. Subsequently, we performed an electrophoretic mobility shift assay (EMSA) and found that the cold probe concentration could abolish the binding activities (Fig. 4c and Supplementary Fig. 14b), confirming that Dt2 could bind to the promoters of *GmAp1a* and *GmAp1d*.

**Dt2-GmAgl22-GmSoc1a could activate the transcription of *GmAp1a* and *GmAp1d***

To further explore how Dt2 affects the transcription of *GmAp1*, we performed transient dual luciferase (Dural-LUC) assay in tobacco leaf system and *Arabidopsis* protoplasts. The results showed that Dt2 functioned as a transcriptional activator to promote the transcription

of *GmAp1a* and *GmAp1d* (Fig. 4d–f and Supplementary Fig. 14c–e), which was consistent with the changes in the expression of *GmAp1* in *Dt2*^OE and *Dt2*^CR lines (Supplementary Fig. 13b). Similarly, we found that Dt2-GmAgl22-GmSoc1a together showed stronger activity than Dt2 alone (Fig. 4g–i and Supplementary Fig. 14f–h).

To check whether *GmAp1* indeed affects the branching of soybean, we overexpressed *GmAp1a* (named *GmAp1a*^OE) and also knocked out the four homologous genes of *GmAp1* (named *GmAp1*^4m) to eliminate their functional redundancy[27]. We found that the *GmAp1a*^OE line exhibited significantly decreased branch number (Fig. 4j–l and Supplementary Fig. 13d), and the quadruple mutant *GmAp1*^4m exhibited a significantly increased branch number (Fig. 4m–o and Supplementary Fig. 13e), confirming that *GmAp1* was indeed involved in branching development in soybean.

**Selection of *Dt2* natural variations under adaptation**

Soybean is a vital crop in China and is planted nationwide from the high-latitude northeast to the low-latitude south. Since branching has a great influence on soybean yield, it has been considered and strongly selected in soybean breeding. Interestingly, we found that the branch number of the soybean accessions from different ecoregions exhibited significant differences: the average branch number of the accessions from higher latitudes was lower than that of accessions from lower latitudes (Fig. 5a).

An investigation of the haplotypes of *Dt2* using our 2898 previously re-sequenced accessions[15] revealed that *Dt2*^HapII did not exist in wild soybean and exhibited an increased ratio from landraces to cultivars (Supplementary Fig. 15a). As *Dt2* is a dominant locus controlling branching in soybean natural population (Fig. 1e, f), we speculated that the natural variation in *Dt2* may be related to the branch number variation in different planting ecoregions. We then investigated the haplotypes of *Dt2* in the cultivated accessions (including landraces and cultivars) and found that the two *Dt2* haplotypes exhibited different geographical distributions: an increased ratio of *Dt2*^HapI/*Dt2*^HapII from

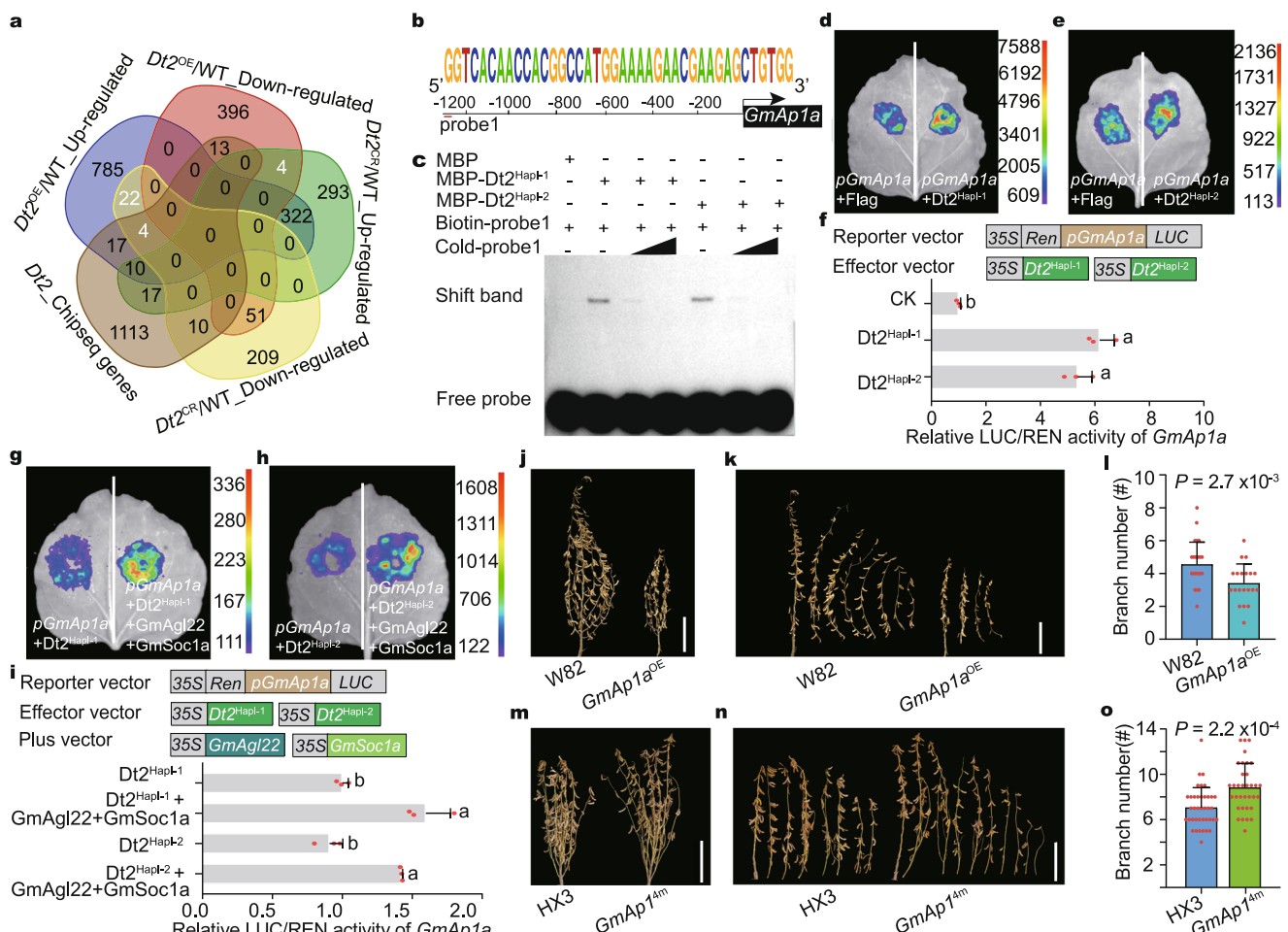

**Fig. 4 | Dt2 directly binds the promoter of GmAp1a and positively regulates its expression. a** Venn diagram among overlapping DEGs from DN50, *Dt2*^CR and *Dt2*^OE lines. **b** A probe sequence that could be bound by the MADS-box protein in the promoter of the *GmAp1a* is predicted in Plantpan (http://plantpan.itps.ncku.edu.tw/). **c** MBP-Dt2^Hapl-1 or MBP-Dt2^Hapl-2 directly bind to the promoter of *GmAp1a* in EMSA. MBP, maltose-binding protein. At least 3 independent replicates are performed and a representative result is shown. **d, e** Transient dual luciferase (dual-LUC) assays of Dt2^Hapl-1 (**d**) and Dt2^Hapl-2 (**e**) binding to the promoter of *GmAp1a* in tobacco leaves. **f** Transient dual luciferase (dual-LUC) assay of Dt2 on the promoter of *GmAp1a* in *Arabidopsis* protoplast (*n* = 3 biologically independent replicates). Different letters indicate statistically significant differences at *P* < 0.05 by one-way ANOVA test. **g, h** Transient dual-LUC assay in tobacco leaves of Dt2^Hapl-1

(**g**) or Dt2^Hapl-2 (**h**) and its interacting proteins GmAgl22 and GmSoc1a on the promoter of *GmAp1a*. **i** Transient dual-LUC assay of Dt2 and its interacting proteins GmAgl22 and GmSoc1a on the promoter of *GmAp1a* in *Arabidopsis* protoplasts (*n* = 3 biologically independent replicates). Different letters indicate statistically significant differences at *P* < 0.05 by one-way ANOVA test. **j, k** Phenotypic comparison between W82 and *GmAp1a*^OE lines. Scale bars, 20 cm. **l** Branch number statistics of W82 and *GmAp1a*^OE lines (*n* = 20 biologically independent plants). Statistical significance is determined using a two-sided *t*-test. **m, n** Phenotypic comparison between the HX3 and *GmAp1*^4m mutation lines in branch number. Scale bars, 20 cm. **o** Branch number statistics of the HX3 and *GmAp1*^4m mutation lines (*n* = 20 biologically independent plants). Data in (**f, i, l** and **o**) are mean ± SEM. Source data are provided as a Source Data file.

higher latitudes to the lower latitudes, which was consistent with the branch number change pattern (Fig. 5a, Supplementary Table 2 and Supplementary Data 1). It has been suggested that the domestication of soybean may have originated in China in the Huanghuai region (ecoregion II in Fig. 5a) and then radiated to the northern and southern regions[28]. $F_{ST}$ analysis showed that the *Dt2* locus exhibited a genetic differentiation tendency between ecoregions II/I, but not between ecoregions II/III (Supplementary Fig. 15b), indicating that the distinct geographic distribution of *Dt2* haplotypes may be related to soybean adaptation to different latitudes.

When a soybean accession from higher latitudes is planted at lower latitudes, it usually exhibits a significant yield decrease due to the early flowering and maturity[29]. The geographic and genetic differentiations of *Dt2* inspired us that modification of *Dt2* may improve the adaptation of soybean. DN50, the parent used for genetic modification of *Dt2* in this study, is an accession from Heilongjiang Province (northeastern of China; 45°77′ N and 126°68′ E), a region located at high latitudes. When DN50 was planted in Beijing (in the middle of

China; 40°22′ N and 116°23′ E) and Hainan (in the southern China, close to the equator; 18°09′ N and 108°48′E), the yields were significantly decreased (Supplementary Table 3). However, the *Dt2*^CR lines showed significantly higher yields than DN50, either at a lower planting density or a higher planting density, which was also associated with its effect on maturity (Supplementary Fig. 16).

## Discussion

Shoot branching is both an agronomically important and a complex developmental trait that can be affected by many factors, of which the transition from the vegetative to reproductive stage is particularly important[30,31]. Functional variation in the genes related to vegetative-reproductive transition tends to cause a coupled phenotypic change in shoot branching morphology and flowering time, as observed, for the *Hd3a* gene in rice[32] and the *VEG1* gene in pea[33]. In this study, we revealed that the dominant gene controlling branch number in soybean natural population, *Dt2*, is a gene related to vegetative-reproductive transition. In addition, the interaction genes and

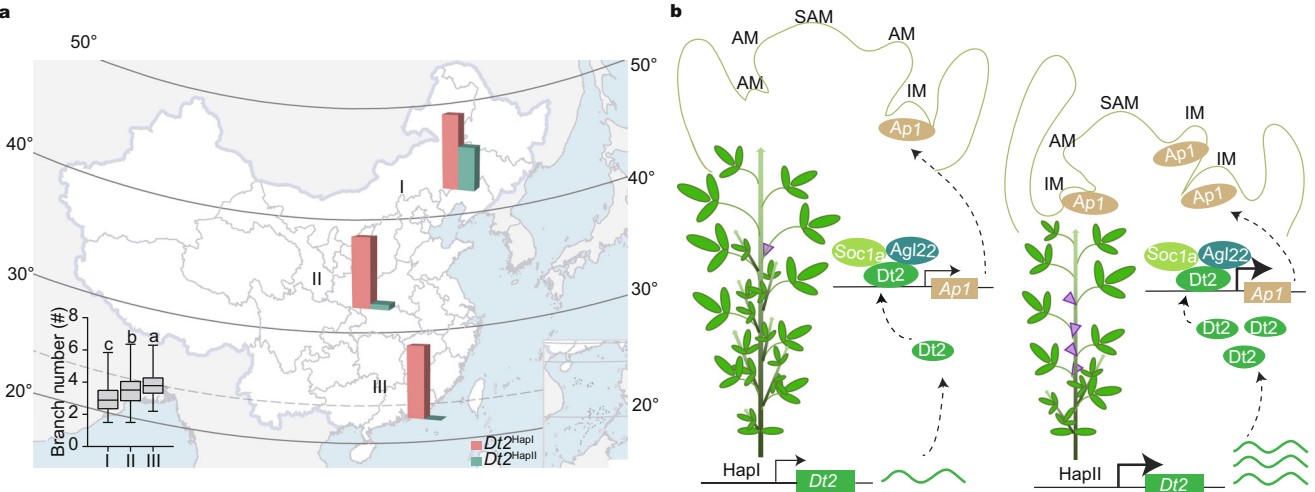

**Fig. 5 | Geographic distribution of different Dt2 haplotypes and a proposed working model of Dt2 in controlling branch number in soybean. a** Geographic distribution of *Dt2* haplotypes in three different ecoregions. The column diagram is the percentage ratio between *Dt2*^HapI and *Dt2*^HapII. The bottom left box plot represents the branch number comparison of accessions in I (*n* = 695 accessions), II (*n* = 821 accessions) and III (*n* = 222 accessions) ecoregions. Data are the mean ± SEM. In box plot (drawn by R 4.1.1 software), the center line indicates the median, the edges of the box represent the first and third quartiles, and the whiskers extend to span a 1.5 interquartile range from the edges. Different letters indicate statistically significant differences at *P* < 0.05 by one-way ANOVA test. **b** Proposed working model of the role of *Dt2* in controlling branch number in soybean. IM inflorescence meristem. AM axillary meristem. SAM shoot apical meristem. The purple triangles represent the transitions from AM to IM.

downstream targets of *Dt2* are also involved in flowering regulation[21–23,27,34]. Therefore, taking into account the abovementioned factors, we proposed a possible working model for the role of *Dt2* in modulating soybean branching (Fig. 5b): *GmAp1* functions as a positive regulator in the transition from vegetative growth into reproductive growth by promoting the transition of axillary meristems (AM) into the inflorescence meristem (IM); Dt2 interacts with GmAgl22 and GmSoc1 to activate the transcription of *GmAp1*; the *Dt2*^HapII promoter has higher transcriptional activity than the *Dt2*^HapI promoter, which in turn enhances the expression of *GmAp1* and promotes earlier transition from vegetative growth to reproductive growth, resulting in earlier flowering and reduced branching. Nevertheless, of the multiple polymorphisms between *Dt2*^HapI and *Dt2*^HapII, we have not disclosed the causal genetic variations responsible for the functional divergence of the two haplotypes. A further investigation of upstream regulatory genes may help us to determine which polymorphisms are essential for the transcription of *Dt2*, which will make the regulatory network more complete.

MADS-box transcription factor genes have been extensively studied in plants and were reported to play important roles in plant development, especially in vegetative-reproductive transition and plant architecture. For instance, *SUPPRESSOR OF OVEREXPRESSION OF CO1* and *FRUITFULL* were found to not only promote flowering, but also affect the determinacy of meristems in *Arabidopsis*[35]. Previous studies in soybean have illuminated that the members involved in the working model of *Dt2* from this study, including *Dt2*, *GmAgl22*, *GmSoc1* and *GmAp1*, all regulate flowering time by being involved in inflorescence transition[18–20,24,27]. Similarly, *MtFUL1-c* and *VEG1/PsFUL1-c*, orthologs of *Dt2* from *Medicago* and pea respectively, were also found to play important roles in the reproductive phase transition[33,36]. Here, we found that these genes not only affect flowering time, but also control branch number in soybean. Consistently, the *Arabidopsis* homologs of *Dt2-GmSoc1a*, *FUL-SOC1*, were also found to regulate branch number[35,37,38]. These results suggested that *Dt2* and other MADS box genes are highly pleiotropic in regulating vegetative-reproductive transition and plant architecture. In addition, these results also indicated that the network of MADS box genes in regulating plant architecture may be functionally conserved among plant species. Taken together previous studies[18–20,24,27], our results revealed that Dt2,

GmAgl22 and GmSoc1 function as a complex to regulate *GmAp1*. Interestingly, the genes involved in the working model of *Dt2* all belong to the MADS-box gene family, indicating a complicated regulatory network among different MADS-box transcription factor genes although they belong to the same transcription factor family. Further investigation of more MADS-box members is needed to reveal more detailed regulatory networks of this important transcription factor gene family.

Soybean was originally domesticated in China and was then introduced into different countries[39]. Because soybean is highly sensitive to photoperiod, selection of the beneficial alleles of the genes that confer adaptation is critical during the spreading process[13,40]. For instance, selection of natural variation in the *J* gene improved the soybean adaptation to the tropical regions and significantly enhanced yield[29], making Brazil one of the largest soybean producing countries today. Here, we determined that the natural variation in *Dt2* is also related to the adaptation of soybean (Fig. 5a, Supplementary Fig. 15), which provides a genetic candidate for adaptation breeding in soybean. In the future, a trial involving a combinational selection/modification of *J* and *Dt2* may enhance soybean adaptation ability.

In addition to the vegetative-reproductive transition, other factors/pathways also determine shoot branching development[1]. Here, we found that the dominant gene associated with branch number in soybean is related to vegetative-reproductive transition. Next, fixing the effect of vegetative-reproductive transition, particularly the effect of the dominant gene *Dt2* may help to identify additional genes functioning in sobyean shoot branching, and will provide more insight for soybean improvement.

## Methods
### Plant materials and growth conditions
The 2409 soybean accessions used for the GWAS were planted at the experimental station of the Tianjin Academy of Agricultural Science, Wuqing (39° 38′ N and 117° 04′ E) during the summer season in 2017 and the Institute of Genetics and Developmental Biology, Chinese Academy of Sciences, Beijing (40° 22′ N and 116° 23′ E) in 2018. At the full-pod stage (R4), 5 representative plants were selected from each accession and the primary effective branch number was identified. The *Dt2*^HapI-1 material was PI 548533 and *Dt2*^HapII was PI 547501.

For the BiFC and dual Luciferase reporter assays, *Arabidopsis* ecotype Col-0 was planted in the greenhouse at 22 °C under an 8 h light and 16 h dark photoperiod. Tobacco (*Nicotiana benthamiana*) was planted in the greenhouse at 22 °C under a 16 h light and 8 h dark photoperiod.

## Vector construction and transformation

To construct the *Dt2* overexpression plasmid, the CDS of *Dt2* was amplified from DN50 and ligated into pTF101, a binary vector containing the CaMV *35 S* promoter and a terminator, with the restriction sites XbaI and SacI. To construct the *GmAgl22* overexpression plasmid, the CDS of *GmAgl22* was ligated into pTF101 with the restriction sites XbaI and SacI. These constructs were introduced into *Agrobacterium tumefaciens* strain EHA101 and then transformed into DN50.

For the CRISPR/Cas9 system experiments, two sgRNAs were designed using Primer Design software. Two *U6* promoters were used for the guide RNA oligonucleotide pair. The *U6* promoter driving a single guide RNA cassette was cloned into the PMDC123 vector[41]. These constructs were introduced into *Agrobacterium tumefaciens* strain EHA105 and then transformed into DN50. The relevant primers used are listed in Supplementary Data 2.

## RNA extraction and expression analysis

Total RNA was extracted using an RNA isolation kit (Tiangen, DP432) according to the manufacturer's protocol and three biological replicates were performed in each experiment. Reverse transcription was performed using a cDNA synthesis kit (Transgen, AE311). Then the cDNA sequence was used as the template for the quantitative real-time PCR. qPCR was performed using LightCycler 480 SYBR Green I Master (Transgen, AQ101-01) on a LightCycler 480 instrument (Roche). Gene expression was normalized to the expression of the soybean gene *ACTIN11*. Fold changes were calculated from the $2^{-\Delta\Delta Ct}$ values. The relevant primers used are listed in Supplementary Data 2.

## GWAS analysis

For GWAS, we used the previously reported SNP dataset[15]. A total of 4,072,231 SNPs were used for association analysis with a minor allele frequency (MAF) of >5% and a missing rate of <10%. GWAS was performed based on a mixed model using the EMMAX software package[42]. EIGENSOFT software[43] was used to perform principal-component analysis of the population, and the first five principal components were included as fixed effects. The matrix of pairwise genetic distances derived from the simple matching coefficients was used as the variance-covariance matrix of the random effects. For the threshold, we defined the whole-genome significance cutoff as the Bonferroni test threshold[44,45], the threshold was set as $-\log(0.05/\text{total SNPs})$, and the genome-wide significance level for branch number was determined as $1 \times 10^{-7.9}$.

## Haplotype analysis of *Dt2* in the soybean population

The SNPs in the 3.5 kb promoter region and full-length genomic region of *Dt2* of 2409 varieties were obtained from the previously reported SNP and INDEL dataset[15]. Then the SNPs were filtered by applying a MAF > 5% cutoff, missing rate <10%, nonfunctional SNP mutation and low association signals SNPs, retaining 35 high-quality SNPs. The association polymorphisms classified the accessions into three major haplotypes.

## Phylogenetic tree analysis

The homology of Dt2 proteins was searched in Phytozome 13 (https://phytozome-next.jgi.doe.gov/blast-search), focusing on genes with homology >50% in soybean, *Arabidopsis* and rice. MEGA 6.0 was used for sequence comparison and phylogenetic tree construction analysis and the bootstrap repetition value was 1000. The phylogenetic tree was further modified by the online tool evolview (https://www.evolgenius.info//evolview/)[46].

## Selection analysis

The genetic differentiation fixation index ($F_{ST}$) was calculated by using VCFtools (0.1.13) with a 20 kb slide window and 2 kb slide step[47]. The first 5% value was used as the threshold in the whole genome.

## In situ hybridization

In situ hybridization treatments were performed as previously described[48,49]. Briefly, the soybean shoot apexes of 10, 16 and 22 day-old seedlings were fixed in 50% formol-acetic-alcohol. Subsequently, the 8 μm-thick SAM samples slices were observed in a conventional light microscope after sample fixation, embedding, sectioning and hybridization. The size of the *Dt2* probe was 228 bp, that of the *GmAgl22* probe was 137 bp, and that of the *GmSoc1a* probe was 140 bp. The primers are listed in Supplementary Data 2.

## RNA-seq sample preparation and sequencing

Lateral buds from the same node of WT and transgenic soybean plants were collected for RNA-seq analysis. Three biological replicates were performed for each sample. The Illumina HiSeq 2000 platform was used to generate 150 bp paired-end reads. And the detailed bioinformatic analyses were performed as previously described[50]. Briefly, the high-quality sequencing reads were mapped to the reference genome with Hisat (v. 2.2.1). And the gene expression counts were calculated using StringTie (v.1.3.4d). The different expression genes analysis were analyzed by the R-edgeR library (https://bioconductor.org/packages/release/bioc/html/edgeR.html).

## Yeast two-hybrid assays

Yeast two-hybrid assays were performed as described in the Yeast Protocols Handbook (Clontech). The coding region sequence of *Dt2*[Hapl-2] was introduced into the prey vector (pGBKT7). To construct the prey vectors, we ligated the full-length CDSs of *GmAgl22* and *GmSoc1a* into the pGADT7 vector. Then pGBKT7-*Dt2*[Hapl-2] was transformed into the Y2HGold strain with pGADT7-*GmAgl22* or pGADT7-*GmSoc1a* and selected on *DDO* (Synthetic Dropout Medium/-Tryptophan-Leucine) and *QDO* (Synthetic Dropout Medium/-Tryptophan-Histone-Leucine-Adenine) media (Clontech). The empty AD or empty BD served as a negative control. The primers are listed in Supplementary Data 2.

## BiFC

For the construction of BiFC vectors, the Gateway-compatible vectors pUGW2-nYFP and pUGW2-cYFP were used to generate vectors in BIFC assays by using Gateway cloning technology. pUGW2-nYFP is the vector for N-terminal fusion to yellow fluorescent protein (nYFP), and pUGW2-cYFP is the vector for C-terminal fusion to YFP (cYFP). The full-length CDS of *Dt2*[Hapl-2] was cloned into pUGW2-nYFP. The full-length CDSs of *GmAgl22* and *GmSoc1a* were cloned into pUGW2-cYFP. *Arabidopsis* protoplasts were prepared for the expression assays. Vectors were co-transformed into *Arabidopsis* protoplasts and incubated at 22 °C in the dark for 12–16 h. YFP fluorescence was visualized using confocal laser scanning microscope (Zeiss LSM 985 NLO).

## Co-IP analysis

Co-IP analysis was performed using *Arabidopsis* protoplasts[47]. To construct the vector, full-length CDSs of *Dt2*[Hapl-1] and *Dt2*[Hapl-2] were cloned into pUC19-*35S*-HA vectors, and the CDSs of *GmAgl22* and *GmSoc1a* were introduced into the pUC19-*35S*-Flag vector. *A. thaliana* protoplasts were transfected with 50 μg of plasmid and incubated overnight under low-light-intensity environment. Total protein was extracted from protoplasts after incubation for 12–16 h using extraction buffer (50 mM Tris-HCl (pH 7.5), 0.5 mM EDTA, 150 mM NaCl, 0.5% np-40, 1 mM PMSF, and 1× complete protease inhibitor cocktail (Roche, 04693132001). The protein lysis product were incubated with Flag magnetic beads (MBL) for 30 min to 1 h and wash the beads four times with a wash buffer that consisted of 50 mM Tris-HCl (pH 7.5),

150 mM NaCl, 20% glycerol, 0.1% Triton X-100, 1 mM EDTA (pH 8.0) and 1× complete protease inhibitor cocktail. The immunoprecipitates were separated using SDS-PAGE and transferred to a nitrocellulose membrane (GE Healthcare). Proteins were detected by treating the membranes with anti-HA (1:5,000, MBL, M180-7) or anti-DDDDK-tag mAb-HRP-DirectT antibodies (1:10,000, MBL, M185-7).

## Transient dual luciferase (dual-LUC) assay

To generate the *pGmAp1a*:LUC and *pGmAp1d*:LUC constructs, we amplified 3-Kb promoter fragments upstream of each gene from Williams 82 and ligated them with the pGreen0800-LUC as the reporter vector. The *p35S-Dt2*[Hapl-1]-Flag, *p35S-Dt2*[Hapl-2]-Flag, *p35S-GmAgl22*-Flag and *p35S-GmSoc1a*-Flag constructs were used as effectors. Transient transactivation assays were performed using *Arabidopsis* protoplasts[51]. In the tobacco leaf system, the promoters of the *GmAp1a* and *GmAp1d* (3000 bp) were cloned into the transient expression vector CP461, which was constructed as the reporter vector. In addition, the *p35S-Dt2*[Hapl-1]-Flag, *p35S-Dt2*[Hapl-2]-Flag, *p35S-GmAgl22*-Flag and *p35S-GmSoc1a*-Flag constructs were used as effectors and these plasmids were transformed into *A. tumefaciens* strain GV3101. Then these strains were injected into tobacco leaves in different combinations with p19, which was used to suppress RNA silencing. Dual luciferase assay reagents (Promega, VPE1910) with the Renilla luciferase gene as an internal control were used for luciferase imaging. The relevant primers are listed in Supplementary Data 2.

## Split luciferase (split-LUC) complementation assay

Luciferase complementation imaging assays were performed as described previously[52]. Briefly, to generate a luciferase complementation vector, pCAMBIA1300-*35S*-NLuc was fused with the C-termini of *Dt2*[Hapl-1], *Dt2*[Hapl-2] and *GmAgl22*, and pCAMBIA1300-*35S*-CLuc was fused with the C-termini of *GmAgl22* and *GmSoc1a*. Transient expression in tobacco (*Nicotiana benthamiana*) leaves was conducted by GV3101 *Agrobacterium* infiltration. Plants were then incubated at 22 °C for 2 days before the LUC activity was measured. Images were captured using the low-light cooled charge-coupled device imaging apparatus NightOWL IILB 983.

## Electrophoretic mobility shift assay

The full length of *Dt2*[Hapl-1] and *Dt2*[Hapl-2] CDSs were amplified and cloned into the Pmal-C5x vector. MBP-Dt2[Hapl-1] and MBP-Dt2[Hapl-2] recombinant proteins with MBP tags were introduced into *Escherichia coli* BL21 (DE3). The recombinant proteins were purified by using maltose resin (NEB, E8021S). DNA probes (Probe1 in *GmAp1a* and Probe2 in *GmAp1d*) were artificially synthesized and labeled with biotin at the 5′ end (Thermo Fisher Scientific). DNA gel shift assays were performed as the protocol described of the LightShift Chemiluminescent EMSA kit (Thermo Fisher Scientific, 20148). The probe sequences are listed in Supplementary Data 2.

## Reporting summary

Further information on research design is available in the Nature Research Reporting Summary linked to this article.

## Data availability

The SNP data of 2409 natural population accessions were reported previously and have been deposited in the Genome Sequence Archive (GSA) database in the BIG Data Center under accession number PRJNA257011, PRJNA394629 and CRA002269 [https://ngdc.cncb.ac.cn/bioproject/browse/PRJCA002030]. The RNA-seq data generated in this study have been deposited into the Genome Sequence Archive (GSA) database in the National Genomics Data Center under accession number SAMC797049-SAMC797057 of PRJCA009434. Requests for materials should be addressed to Z.T. Source data are provided with this paper.

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

## Acknowledgements

This work was supported by the National Key Research and Development Program of China (grant no. 2021YFF1001201), National Natural Science Foundation of China (grant no. 31788103, 32090064), Hainan Yazhou Bay Seed Laboratory Project (grant no. B21HJ0002), and "Strategic Priority Research Program" of the Chinese Academy of Sciences (grant no. XDA24030501).

## Author contributions

Z.T. designed and supervised the project; Q.L., L.C., X.Y., H.Y., K.K., L.F., Z.D., Y.Y., S. Liang, X.L., G.Z., and M.Z. performed the experiments; Q.L., S. Liu, Z.Z., Y.L., F.K., and Z.T. analyzed the data; Q.L. and Z.T. wrote the paper.

## Competing interests

The authors declare no competing interests.

## Additional information

**Correspondence and requests** for materials should be addressed to Fanjiang Kong or Zhixi Tian.

