## [Peer Review File · Nature Communications]

Natural variation of Dt2 determines branching in soybeanReviewers' Comments:

Reviewer #1:

Remarks to the Author:

This study addresses the phenotypic and molecular characterization of a well known soybean gene, Dt2, that has been previously identified by its large effects on growth habit and plant architecture. This study provides new details on Dt2 phenotypic effects on the number of branches, as well as on the molecular mechanisms of action for this MADS transcription factor. Authors show that Dt2 physically interact with two other MADS transcription factors, GmAgl22 and GmSoc, and bind to the promoters of GmAp1a and GmAp1d. Overall, these results reinforce and refine the molecular function previously established for Dt2 (Ping et al., 2014; Liu et al., 2016; Zhang et al., 2019). In addition, this study identifies Dt2 as a gene contributing to the natural variation for plant architecture and yield in soybean, in agreement with previous studies describing the effect of Dt2 natural gain-of-function alleles affecting growth habit (Ping et al., 2014). Genetic and molecular biology experiments are carefully performed, and the manuscript is in general written in a concise and comprehensive manner.

However, there are two major conclusions of this study that are not well supported by the data presented.

1. In the section of results entitled "Selection of Dt2 natural variation under adaptation" (lines 218-243), as well as in introduction and abstract, authors speculate on selection on Dt2 gene, and they conclude that "Modulation of Dt2 enhanced soybean adaptation and yield" (abstract, line 22). However, that section of results only shows: i) that the two major haplotypes of Dt2 display different geographic distribution; ii) and that the yield effects of Dt2 artificial genetic variation (Dt2CR mutants, functionally different from natural variants) depend on the environment (significant genotype by environment interaction). Both results are commonplace for natural variants of many genes, but do not indicate selection or adaptation, which are big words from an evolutionary point of view. Demonstrating that Dt2 natural allelic variation is involved in soybean selection/adaptation requires separating demographical/historical effects on our traits of interest (genomic background), from the advantageous or adaptive effects of specific natural alleles at Dt2, which is not considered in this study. Furthermore, as shown in a previous study (ref 18 of the manuscript) HapII haplotype is probably a gain-of-function allele recently derived (such derived state and its age estimate need to be inferred). Therefore, addressing the adaptive relevance of the natural variation at Dt2 should focus in understanding if this new derived allele is adapted to particular environmental conditions (local/regional adaptation) in northern latitudes, or if, on the contrary, it might be neutral (it might show conditional neutrality). Authors do not take into account this allele differences and focus their speculation on the adaptive value of the major ancestral allele.

2. Authors mostly describe Dt2 as a regulator of branching in soybean. However, it has been well established in references cited in the manuscript (e.g. refs 17, 18, 19) that Dt2 is a highly pleiotropic gene because it affects the entire soybean plant architecture, and subsequently yield. As shown in this study, the effects of Dt2 natural variants on soybean yield are determined not only by effects on the number of branches, but also on flowering time, the length of the stems, the number of nodes, etc. More importantly, in agreement with studies in this and other plant species, Dt2 and other MADS box genes act upstream in the network regulating plant architecture, but they show no specific/direct regulation on axillary bud development and dormancy or branching. Therefore, it seems more rigorous and informative to describe the contribution of Dt2 as a plant architecture gene and not as a branch regulator.

Minor comments:

- The molecular mechanisms of Dt2 action described here and in previous studies, together with the effects of Dt2 natural variants, show a large functional conservation among distant plant species for the network of MADS box genes regulating plant architecture. It might be interesting to briefly compare the results of this study with the analogous network described in other plant systems.
- In the first section of Results (lines 66-67) authors state: "Phylogenetic and principal component

analysis divided the studied accessions into two major groups: landrace and cultivar (Supplementary Fig. 1a, b)". Such Figures do not show significant genetic differentiation between the two groups of soybean strains, but those groups seem defined a priori.

- In the last section of results, lines 222-223, the sentence "we found that the branch number of the soybean accessions from different ecotypes exhibited significant differences" is very unclear. It is unknown if accessions refer to landraces or/and cultivars. In addition, the term "ecotypes" in this sentence and in line 227 seems inappropriate or confusing.
- According to my calculation, the Bonferroni threshold for the 4 million SNPs used in GWAS analyses should be $1 \times 10^{-7.9}$. However, authors state in line 329 that they used a Bonferroni cut-off of $1 \times 10^{-6.5}$. Did they use a different than Bonferroni correction for multiple testing?
- In line 86, where it says "loci" should better read "SNPs".
- In line 156, "their transgenic parents" might better be referred to as "non-transgenic parents".

Some grammar/typographical errors:

- line 176, change "met" with "meeting".
- line 181, gene "SoyZH13_01G63000" appears as "SoyZH13_01G60300" in Supplemental Figure 12a.
- lines 187, 688, 689, "in situ hybrid assay" should better read "in situ hybridization assay".
- line 197, the word "respectively" seems incorrect (remove?)
- line 211, the word "simultaneously" is misleading. I suggest replacing with also.
- line 323, where it reads "were" should read "was".
- line 597, where it says "their" should say "its".
- According to Supplemental Figure 11a, in line 693, where it reads "Dt2OE" and "Dt2OE/WT" should read "Dt2CR" and "Dt2CR/WT", respectively.
- According to Supplemental Figure 11b, in line 694, where it reads "Dt2CR" and "Dt2CR/WT" should say "Dt2OE" and "Dt2OE/WT", respectively.

Reviewer #2:

Remarks to the Author:

In the manuscript 'Natural variation of Dt2 determines branching and enhances yield in soybean', Liang et al. reveal that the natural variation in soybean branch number can be explained by differences at the Dt2 locus. Analysis of two haplotypes with extreme phenotypes and of CRISPR mutants and overexpression lines shows that higher expression of Dt2 is linked to a lower branch number, while lower expression results in a higher branch number. Thus, Dt2 is negatively regulating branch number. The manuscript contains interesting new insights into the role of Dt2 in soybean branching, while previous work mainly focused on the role of Dt2 in the regulation of inflorescence determinacy. However, several things that the authors present could have been expected based on earlier work. For example:

- The Arabidopsis homologs of Dt2-Soc1a, FUL-SOC1, regulate branch number (Melzer et al., 2008; Bemer et al., 2017; Karami et al., 2020)
- Dt2 interacts with Soc1a and can upregulate GmAp1a to regulate (lateral) flowering time (Liu et al., 2016; Zhang et al., 2019)
- Soc1a regulates stem node number (Kou et al., 2022)
- Dt2 orthologs MtFUL1-c and VEG1/PsFUL1-c function in the transition to the reproductive phase, also in axillary shoots (e.g. Cheng et al., 2018)

Thus, the authors present a well-thought approach to identify the locus/gene that underlies the difference in branch number between two haplotypes from different latitudes. In general, the experiments seem solid, the analyses thorough, and the manuscript is very well written. The identification of Dt2 is not very surprising however, given the already available data. The subsequent downstream analysis is not very thorough and it remains for example unclear which difference at the Dt2 locus of the two contrasting haplotypes brings about the difference in expression and phenotype.

Additional comments

- Results, line 80, FRUITFUL should be FRUITFULL. However, actually, Dt2 belongs to the euFULII-clade of the AP1/FUL subfamily and is thereby a closer homolog of AGL79 (Arabidopsis)/VEG1(pea)/MBP20 (tomato).
- The phenotypic description is not clear to me. Are more branches/axillary nodes formed or is it a matter of outgrowth of existing meristems/primordia? Could you consider the axillary meristem 'dormant' as long as the transition to IM has not been made and is it growing out when this transition has occurred? This should be described/investigated better, also in the light of the model that the authors present in Figure 5.
- It is odd that the expression of Dt2 is non-detectable in the CRISPR lines. In particular given the 1 nt deletion in the CR-1 line. These mutations lead to impaired protein function, but usually not to a decrease in expression. It is probably that Dt2 upregulates its own expression, but a more mild down-regulation would be expected in that case.
- The CARG-box motif prediction in Suppl. Fig. 13a does not look very accurate, as a CARG-box is more than only the C at position 1 and G at position 10. The quality of the EMSA is also not very high (or the motif does not contain a good CARG-box).
- It is a pity that there is only an overexpression line for Agl22, as MADS-domain TFs can all bind to CARG-boxes and the phenotype obtained in an overexpression line does not prove that the specific MADS-box gene actually functions in the regulation of branching number. However, this is likely based on the interaction of Agl22 with SOC1a/Dt2 and its presence in the axillary bud.
- The link to Arabidopsis research concerning the role of FUL-like genes in branching has not been described, nor the role of VEG1/MtFUL1-c in other legumes (in the Introduction or Discussion section).

Reviewer #3:

Remarks to the Author:

The authors presented a mature and very well-written manuscript showcasing how natural variation at Dt2 determines shoot branching and yield in soybeans. Using genetic, molecular, biochemical, and transgenic approaches authors convincingly show that Dt2 worked by interacting with GmAgl22 and GmSoc1a to physically bind to the promoter of GmAp1a and GmAp1d and to activate their transcription. Population genetic investigations showed that selection at Dt2 was associated with geographic differentiation.

This work is highly original and reflects how few we actually know about plant architectural mechanisms in one of our major legume crops. Hence, I enjoyed reading this manuscript. I don't have considerable objections for this study. However, I still have one point for consideration and improvement:

1) I found the model authors derived in figure 5b quite appealing; however, it is not very clear which color the reproductive structures have. As far as I could comprehend AxM turned to IMs are indicated by pinkish triangles. If so, then taller, more indeterminate plants have less reproductive structures than smaller ones, which is in contrast to the actual data. Please clarify the figure in a way that it becomes consistent with your findings.

Reviewer #4:

Remarks to the Author:

The objective of Liang et al. was to identify key genes controlling branch number in soybean, which is an important shoot architecture trait related to seed yield. The authors performed a GWAS on a panel of 2573 accessions that had been re-sequenced, providing over 4M SNPs. The only association found between branch number of SNPs was in the region of Dt2. The authors performed gene expression analyses and found that gene expression at Dt2 was negatively correlated with branch number. They went further creating gene edited knockouts and transgenic overexpressed lines at Dt2 and interacting

genes, showing that Dt2 and genes producing downstream products affect branch number in soybean. The results from this study show that Dt2 has effects beyond determinancy, affecting shoot architecture in addition to timing of vegetative to reproductive phases.

Overall, I think this is a terrific study that used many tools and thoroughly showed the important effect of Dt2. Quantitative genetics combined with functional analyses through genetic modification and gene expression analyses were deployed.

Lines 28-43: Overall, the tone of this paragraph makes it sound like these advances were on the basis of design using knowledge about genotype, or basically that phenotype followed selected on genotype, when in fact many of these advances were the product of unconscious selection. In other words, changes in the genotype followed selections on the phenotype.

Line 41: Need reference backing up statement that new superior cultivars were developed by introducing ipa1.

Lines 72-73: Here it is claimed there is evidence that gene 18g_242900 showed higher specific expression among 16 possible candidate genes in the region identified by GWAS, but in Fig S2, this is not apparent as this gene does not seem to have a expressions profile unique at all relative to the others. Also, Fig 1 is cited here as supported evidence in expression but this data is not provided in Figure 1.

Line 82: Why deploy a MAF cutoff of 0.1 here and not 0.05 like described in the Methods section? In general, with over 2500 accessions, a much lower MAF than 0.05 can be used in general. I think this cutoffs should be justified or altered as they can affect results and are quite arbitrary.

Line 242: Here and elsewhere I think there needs to be more acknowledgment of the effect of maturity on yield with changed in Dt2. Many so-called yield genes just affect maturity, and it is easy to breed for increased yield by increasing maturity, but that is not practical as often a limited growing season is reality.

REVIEWER COMMENTS

Reviewer #1 (Remarks to the Author):

This study addresses the phenotypic and molecular characterization of a well-known soybean gene, Dt2, that has been previously identified by its large effects on growth habit and plant architecture. This study provides new details on Dt2 phenotypic effects on the number of branches, as well as on the molecular mechanisms of action for this MADS transcription factor. Authors show that Dt2 physically interact with two other MADS transcription factors, GmAgl22 and GmSoc, and bind to the promoters of GmAp1a and GmAp1d. Overall, these results reinforce and refine the molecular function previously established for Dt2 (Ping et al., 2014; Liu et al., 2016; Zhang et al., 2019). In addition, this study identifies Dt2 as a gene contributing to the natural variation for plant architecture and yield in soybean, in agreement with previous studies describing the effect of Dt2 natural gain-of-function alleles affecting growth habit (Ping et al., 2014). Genetic and molecular biology experiments are carefully performed, and the manuscript is in general written in a concise and comprehensive manner.

Response: Thank you so much for the encouragement and your constructive comments. We have tried our best to revise the manuscript, and we hope it will address all your questions. Thank you again.

However, there are two major conclusions of this study that are not well supported by the data presented.

1. In the section of results entitled “Selection of Dt2 natural variation under adaptation” (lines 218-243), as well as in introduction and abstract, authors speculate on selection on Dt2 gene, and they conclude that “Modulation of Dt2 enhanced soybean adaptation and yield” (abstract, line 22). However, that section of results only shows: i) that the two major haplotypes of Dt2 display different geographic distribution; ii) and that the yield effects of Dt2 artificial genetic variation (Dt2CR mutants, functionally different from natural variants) depend on the environment (significant genotype by environment interaction). Both results are commonplace for natural variants of many genes, but do not indicate selection or adaptation, which are big words from an evolutionary point of view. Demonstrating that Dt2 natural allelic variation is involved in soybean

selection/adaptation requires separating demographical/historical effects on our traits of interest (genomic background), from the advantageous or adaptive effects of specific natural alleles at *Dt2*, which is not considered in this study. Furthermore, as shown in a previous study (ref 18 of the manuscript) HapII haplotype is probably a gain-of-function allele recently derived (such derived state and its age estimate need to be inferred). Therefore, addressing the adaptive relevance of the natural variation at *Dt2* should focus in understanding if this new derived allele is adapted to particular environmental conditions (local/regional adaptation) in northern latitudes, or if, on the contrary, it might be neutral (it might show conditional neutrality). Authors do not take into account this allele differences and focus their speculation on the adaptive value of the major ancestral allele.

Response: Thank you so much for the important and valuable comments.

Following your direction, we performed additional analyses, including 1) haplotype analysis of the *Dt2* haplotypes in a re-sequenced natural population of 2989 soybean accessions to investigate the evolutionary history of different haplotypes, and 2) selection analysis of the *Dt2* haplotypes among different geographic regions to check if the haplotypes exhibited sectional characters between different regions (**Supplementary Fig. 15**). It has been suggested that the domestication of soybean may have occurred in China in the Huanghuai region (Region II in Figure 5a) and then radiated to the northern and southern regions¹. Our new analyses showed that *Dt2*^{HapII} (SNP from the promoter) does not exist in wild soybeans (**Supplementary Fig. 15a**). The selection analysis indicated that selection for *Dt2*^{HapII} may have occurred during the soybean radiation from the Huanghuai region to the northern region (Region II to Region I); however, no selection character was observed during radiation from the Huanghuai region to the southern region (Region II to Region III) (**Supplementary Fig. 15b**). Therefore, the results indicated that the selection of *Dt2* may contribute to adaptation during soybean radiation from the Huanghuai region to the northern region. Nevertheless, as you mentioned in the next question, because *Dt2* is a highly pleiotropic gene, adaptation may be a result of the selection of multiple traits in addition to the selection of branching number.

The section “Selection of *Dt2* natural variation under adaptation” was revised accordingly: “An investigation of the haplotypes of *Dt2* using our 2,898 previously re-

sequenced accessions¹⁵ revealed that *Dt2*^{HapII} did not exist in wild soybean and exhibited an increased ratio from landraces to cultivars (**Supplementary Fig. 15a**). As *Dt2* is a dominant locus controlling branching in soybean natural population (**Fig. 1e, f**), we speculated that the natural variation in *Dt2* may be related to the branch number variation in different planting ecoregions. We then investigated the haplotypes of *Dt2* in the cultivated accessions (including landraces and cultivars) and found that the two *Dt2* haplotypes exhibited different geographical distributions: an increased ratio of *Dt2*^{HapI}/*Dt2*^{HapII} from higher latitudes to the lower latitudes, which was consistent with the branch number change pattern (**Fig. 5a, Supplementary Fig. 16a and Supplementary Table 3**). It has been suggested that the domestication of soybean may have originated in China in the Huanghuai region (ecoregion II in Figure 5a) and then radiated to the northern and southern regions²⁹. *F_{ST}* analysis showed that the *Dt2* locus exhibited a selection tendency between ecoregions II/I, but not between ecoregions II/III (**Supplementary Fig. 15b**), indicating that the differential distribution of *Dt2* haplotypes may be related to soybean adaptation to different latitudes.” (Lines 227-242).

We also realized that the conclusion “Modulation of *Dt2* enhanced soybean adaptation and yield” was not proper although *Dt2*^{CR} lines could enhance soybean yield under some environments, which may still be related to its effect on flowering time, branching number and other traits. Therefore, we deleted the sentence “Modulation of *Dt2* enhanced soybean adaptation and yield” from the abstract in the revised manuscript. Accordingly, we also changed the title “Natural variation of *Dt2* determines branching and enhances yield in soybean” to “Natural variation of *Dt2* determines branching in soybean”.

Thank you again.

2. Authors mostly describe *Dt2* as a regulator of branching in soybean. However, it has been well established in references cited in the manuscript (e.g. refs 17, 18, 19) that *Dt2* is a highly pleiotropic gene because it affects the entire soybean plant architecture, and subsequently yield. As shown in this study, the effects of *Dt2* natural variants on soybean yield are determined not only by effects on the number of branches, but also on flowering time, the length of the stems, the number of nodes, etc. More importantly, in agreement with studies in this and other plant species, *Dt2* and other MADS box

genes act upstream in the network regulating plant architecture, but they show no specific/direct regulation on axillary bud development and dormancy or branching. Therefore, it seems more rigorous and informative to describe the contribution of *Dt2* as a plant architecture gene and not as a branch regulator.

Response: We agree with you that *Dt2* is a highly pleiotropic gene. Yes, based on previous observations of the effects of *Dt2* on flowering time, the length of the stems, the number of nodes, plusing the new finding of its effect on the number of branches from this study, it is more rigorous and informative to describe the contribution of *Dt2* as a plant architecture gene than only to say it as a branch regulator. Following your comments, we have revised the description in the discussion: “These results suggested that *Dt2* and other MADS box genes are highly pleiotropic in regulating vegetative-reproductive transition and plant architecture” (Lines 290-292).

Minor comments:

- The molecular mechanisms of *Dt2* action described here and in previous studies, together with the effects of *Dt2* natural variants, show a large functional conservation among distant plant species for the network of MADS box genes regulating plant architecture. It might be interesting to briefly compare the results of this study with the analogous network described in other plant systems.

Response: It is a very good suggestion to make a brief comparison functional conservation among distant plant species for the network of MADS box genes. In the revised manuscript, we added a statistical analysis for the growth state of axillary buds in leaf axils. We found that the effect of *Dt2* on branch development may be related to inflorescence determinacy (Supplementary Fig. 7), which further indicated the functional conservation of this gene family in different species. This part was added in to the section “Functional validation of the role of *Dt2* in controlling branch number ”: “Growth state statistics of axillary buds in leaf axils between DN50 and *Dt2*^{CR} lines showed that the effect of *Dt2* on branch development may be related to inflorescence determinacy (**Supplementary Fig. 7**)” (Lines 124-127).

As you suggested, we also made a brief comparison and discussion in the discussion section: “MADS-box transcription factor genes have been extensively

studied in plants and were reported to play important roles in plant development, especially in vegetative-reproductive transition and plant architecture. For instance, *SUPPRESSOR OF OVEREXPRESSION OF CO1* and *FRUITFULL* were found to not only promote flowering, but also affect the determinacy of meristems in *Arabidopsis*³⁶. Previous studies in soybean have illuminated that the members involved in the working model of *Dt2* from this study, including *Dt2*, *GmAg122*, *GmSoc1* and *GmAp1*, all regulate flowering time by being involved in inflorescence transition^{18-20,25,28}. Similarly, *MtFUL1-c* and *VEG1/PsFUL1-c*, orthologs of *Dt2* from *Medicago* and pea respectively, were also found to play important roles in the reproductive phase transition^{34,37}. Here, we found that these genes not only affect flowering time, but also control branch number in soybean. Consistently, the *Arabidopsis* homologs of *Dt2-GmSoc1a*, *FUL-SOC1*, were also found to regulate branch number^{36,38,39}. These results suggested that *Dt2* and other MADS box genes are highly pleiotropic in regulating vegetative-reproductive transition and plant architecture. In addition, these results also indicated that the network of MADS box genes in regulating plant architecture may be functionally conserved among plant species. Taken together previous studies^{18-20,25,28}, our results revealed that *Dt2*, *GmAg122* and *GmSoc1* function as a complex to regulate *GmAp1*. Interestingly, the genes involved in the working model of *Dt2* all belong to the MADS-box gene family, indicating a complicated regulatory network among different MADS-box transcription factor genes although they belong to the same transcription factor family. Further investigation of more MADS-box members is needed to reveal more detailed regulatory networks of this important transcription factor gene family.”
(Lines 278-300).

- In the first section of Results (lines 66-67) authors state: “Phylogenetic and principal component analysis divided the studied accessions into two major groups: landrace and cultivar (Supplementary Fig. 1a, b)”. Such Figures do not show significant genetic differentiation between the two groups of soybean strains, but those groups seem defined a priori.

Response: Thank you so much for pointing out this problem. We apologize so much that we did not clarify it correctly. The population we used for the GWAS included landraces and cultivars, which were classified based on breeding history. As you pointed out the phylogenetic and PCA analyses showed that the landraces and cultivars do not show significant genetic differentiation, which is consistent with previous studies^{2,13,14}. In addition, the non-significant genetic differentiation between landraces and cultivars will make the GWAS robust because no significant population structure exists. We have corrected the clarification as follows: “Phylogenetic and principal component analysis of the landraces and cultivars did not show significant genetic differentiation (Supplementary Fig. 1a, b)” (Lines 65-67).

- In the last section of results, lines 222-223, the sentence “we found that the branch number of the soybean accessions from different ecotypes exhibited significant differences” is very unclear. It is unknown if accessions refer to landraces or/and cultivars. In addition, the term “ecotypes” in this sentence and in line 227 seems inappropriate or confusing.

Response: Thank you so much for the careful review. We are so sorry that we did not clarify it clearly previously. The accessions refer to both landraces and cultivars. The accession information was also listed in Supplementary Table 3. In addition, yes, as you mentioned, the term “ecotypes” is unclear and confusing. We have revised the term to “ecoregions”. This part was revised as follows: “Interestingly, we found that the branch number of the soybean accessions from different ecoregion exhibited significant differences: the average branch number of the accessions from higher latitudes was lower than that of accessions from lower latitudes (Fig. 5a).

As *Dt2* is a dominant locus controlling branching in soybean natural populations (Fig. 1e, f), we speculated that the natural variation in *Dt2* may be related to the branch number variation in different planting ecoregions. We then investigated the haplotypes of the accessions (including landraces and cultivars) and found that the two *Dt2* haplotypes exhibited different geographical distributions: an increased ratio of $Dt2^{\text{HapI}}/Dt2^{\text{HapII}}$ from higher latitudes to lower latitudes (Fig. 5a, Supplementary Fig. 14a and Supplementary Table 3).” (Lines 223-226, 229-237).

Thank you again.

- According to my calculation, the Bonferroni threshold for the 4 million SNPS used in GWAS analyses should be $1 \times 10^{-7.9}$. However, authors state in line 329 that they used a Bonferroni cut-off of $1 \times 10^{-6.5}$. $-\log(1/\text{snp})$. Did they use a different than Bonferroni correction for multiple testing?

Response: Thank you so much for pointing out this problem. Following your suggestion, we reviewed the calculation and found that we incorrectly used a parameter in the Bonferroni threshold calculation formula. We used $-\log(1/\text{Total SNPs})$ instead of $-\log(0.05/\text{Total SNPs})$. Following your direction, we recalculated the Bonferroni threshold using the formula of $-\log(0.05/\text{Total SNPs})$ and determined that the significance cutoff P value is $1 \times 10^{-7.9}$, which is exactly the same as yours. We had corrected this in Fig. 1a and Supplementary Fig.1d, e and Supplementary Fig.2b. We also revised the text accordingly.

“GWAS performed using a mixed linear model revealed a stable association signal across the two years in a 40-Kb interval block on chromosome 18 (**Fig. 1a-d** and **Supplementary Fig. 1d-g**). Within this 40-Kb interval, a total of 5 protein-coding genes were annotated according to the reference genome ZH13¹⁵” (Lines 67-71).

“For the threshold, we defined the whole-genome significance cutoff as the Bonferroni test threshold^{16,17}, the threshold was set as $-\log(0.05/\text{total SNPs})$, and the genome-wide significance level for branch number was determined as $1 \times 10^{-7.9}$ ” (Lines 362-364).

We sincerely apologize for the mistake and sincerely appreciate your careful review, which allowed us to correct this mistake.

- In line 86, where it says “loci” should better read “SNPs”.

Response: It has been revised as you suggested.

- In line 156, “their transgenic parents” might better be referred to as “non-transgenic parents”.

Response: It has been revised as you suggested.

Some grammar/typographical errors:

- line 176, change “met” with “meeting”.

Response: It has been revised as you suggested.

- line 181, gene “SoyZH13_01G63000” appears as “SoyZH13_01G60300” in Supplemental Figure 12a.

Response: We sincerely apologize for this mistake. This has been corrected. Thank you so much.

- lines 187, 688, 689, “in situ hybrid assay” should better read “in situ hybridization assay”.

Response: “in situ hybrid assay” has been revised to “in situ hybridization assay” throughout the entire text.

- line 197, the word “respectively” seems incorrect (remove?)

Response: Yes, the word is not necessary. We has deleted this text as you suggested. Thank you.

- line 211, the word “simultaneously” is misleading. I suggest replacing with also.

Response: “simultaneously” has been replaced with “also” as you suggested.

- line 323, where it reads “were” should read “was”.

Response: It has been revised as you suggested.

- line 597, where it says “their” should say “its”.

Response: It has been revised as you suggested.

- According to Supplemental Figure 11a, in line 693, where it reads “Dt2^{OE}” and “Dt2^{OE}/WT“ should read “Dt2^{CR}” and “Dt2^{CR}/WT“, respectively.

Response: We sincerely apologize for the mistakes. They were corrected. Thank you so much.

- According to Supplemental Figure 11b, in line 694, where it reads “Dt2^{CR}” and “Dt2^{CR}/WT“ should say “Dt2^{OE}” and “Dt2^{OE}/WT“, respectively.

Response: We sincerely apologize for the mistakes. They were corrected. Thank you so much

Reviewer #2 (Remarks to the Author):

In the manuscript ‘Natural variation of *Dt2* determines branching and enhances yield in soybean’, Liang et al. reveal that the natural variation in soybean branch number can be explained by differences at the *Dt2* locus. Analysis of two haplotypes with extreme phenotypes and of CRISPR mutants and overexpression lines shows that higher expression of *Dt2* is linked to a lower branch number, while lower expression results in a higher branch number. Thus, *Dt2* is negatively regulating branch number. The manuscript contains interesting new insights into the role of *Dt2* in soybean branching, while previous work mainly focused on the role of *Dt2* in the regulation of inflorescence determinacy. However, several things that the authors present could have been expected based on earlier work. For example:

- The Arabidopsis homologs of *Dt2*-*Soc1a*, *FUL*-*SOC1*, regulate branch number (Melzer et al., 2008; Bemer et al., 2017; Karami et al., 2020)
- *Dt2* interacts with *Soc1a* and can upregulate *GmAp1a* to regulate (lateral) flowering time (Liu et al., 2016; Zhang et al., 2019)
- *Soc1a* regulates stem node number (Kou et al., 2022)
- *Dt2* orthologs *MtFUL1-c* and *VEG1/PsFUL1-c* function in the transition to the reproductive phase, also in axillary shoots (e.g. Cheng et al., 2018)

Thus, the authors present a well-thought approach to identify the locus/gene that underlies the difference in branch number between two haplotypes from different latitudes. In general, the experiments seem solid, the analyses thorough, and the manuscript is very well written. The identification of *Dt2* is not very surprising however, given the already available data. The subsequent downstream analysis is not very thorough and it remains for example unclear which difference at the *Dt2* locus of the two contrasting haplotypes brings about the difference in expression and phenotype.

Response: Thank you for your encouragement and helpful suggestions.

Yes, as you mentioned, previous work on *Dt2* mainly focused on its role in the regulation of inflorescence determinacy, which has been referred in our manuscript. Nevertheless, the effect of *Dt2* on branching has not been reported. Moreover, other

genes determining the soybean branching in the natural population have not been reported neither. We know that branching is a very important agronomic trait related to seed yield for soybean. Therefore, the purpose of this study was to identify the genes controlling branch number in the soybean natural population. Luckily, we identified a predominant association locus from the GWAS. Unfortunately, the causal gene is a previously reported well-known gene, *Dt2*. Nevertheless, we believe the result is still an important contribution: 1) we identified the major determinant of branch number in soybean natural population (although it is *Dt2*), which provides important and valuable information for the fundamental study and breeding society of soybean; 2) we established a preliminary working model of *Dt2* in controlling branching in soybean by functionally validating the interaction proteins and downstream regulatory genes of *Dt2*; 3) we found that the genetic variations in the promoter play important in gene expression regulation, which has never been reported before.

Therefore, as you said, although the gene we identified is a previously reported gene, we provided interesting new insights into the role of *Dt2* in soybean branching. The results also illuminated the importance of *Dt2* in soybean agronomic trait determination and breeding. We also agree with you that there are still unsolved questions in this study, such as how the SNPs on the promoter result in the difference in gene expression level. We determined that the promoters from the different haplotypes showed differences in transcriptional activity (Fig. 1g). However, many more explorations are needed to clearly clarify the regulatory mechanism, which may be another independent project. We also discussed in the discussion section: “Nevertheless, of the multiple polymorphisms between *Dt2*^{HapI} and *Dt2*^{HapII}, we have not disclosed the causal genetic variations responsible for the functional divergence of the two haplotypes. A further investigation of upstream regulatory genes may help us to determine which polymorphisms are essential for the transcription of *Dt2*, which will make the regulatory module more complete”. We appreciate your suggestions, and will work hard to elucidate the mechanism. We hope we can report additional exciting discoveries soon. Thank you so much.

Additional comments

- Results, line 80, FRUITFUL should be FRUITFULL. However, actually, Dt2 belongs

to the euFULII-clade of the AP1/FUL subfamily and is thereby a closer homolog of AGL79 (*Arabidopsis*)/VEG1(pea)/MBP20 (tomato).

Response: Thank you so much for this important comment. We have corrected the text as you suggested: “Phylogenetic analysis found that *Dt2* belongs to the MADS-box transcription factor family and shares high homology with *AGL79* in *Arabidopsis*, a member of the AP1/FUL subfamily (**Supplementary Fig. 2c**).” (Lines 78-80).

- The phenotypic description is not clear to me. Are more branches/axillary nodes formed or is it a matter of outgrowth of existing meristems/primordia? Could you consider the axillary meristem ‘dormant’ as long as the transition to IM has not been made and is it growing out when this transition has occurred? This should be described/investigated better, also in the light of the model that the authors present in Figure 5.

Response: Thank you for your question. To observe the branching phenotype more clearly, we statistically analyzed the number of axillary buds and the growth stage (to remain dormant or to undergo outgrowth) in the whole stage between the DN50 and *Dt2*^{CR} lines. We found that the *Dt2*^{CR} lines had a higher ratio of dormant buds axillary buds than DN50. In addition, we also found that the effect of *Dt2* on branch development may be related to the inflorescence determinacy. The new results were added to the section “Natural variation in *Dt2* predominantly determines soybean branching”: “Growth state statistics of axillary buds in leaf axils between DN50 and *Dt2*^{CR} lines showed that the effect of *Dt2* on branch development may be related to inflorescence determinacy (**Supplementary Fig. 7**).” (Lines 124-127).

- It is odd that the expression of *Dt2* is non-detectable in the CRISPR lines. In particular given the 1 nt deletion in the CR-1 line. These mutations lead to impaired protein function, but usually not to a decrease in expression. It is probably that *Dt2* upregulates its own expression, but a more mild down-regulation would be expected in that case.

Response: Thank you for pointing out this question. We are also curious about these results, although this kind of observation was also found before^{18,19}. For example, the *IPAI* or *D53* expression level significantly was downregulated in *ipal-10d* or *d53*

mutant¹⁹. One possibility is, as you suggested, that Dt2 is involved in the regulation of its own expression. Another possibility is nonsense-mediated mRNA decay (NMD), which is a eukaryotic surveillance process that promotes selective degradation of imperfect messages containing premature translation termination codons²⁰. You have given us a very good suggestion and direction to further elucidate the regulatory mechanism of *Dt2*, which deserves to be determined in the future. Thank you.

- The CARG-box motif prediction in Suppl. Fig. 13a does not look very accurate, as a CARG-box is more than only the C at position 1 and G at position 10.

Response: Thank you so much for this important comment. Yes, as you pointed out, CARG-box motifs have more than just the C at position 1 and G at position 10. In the last version, we only showed the target sequence of the MADS-box protein which was predicted by the PlantPan. Referring to a previous report from the Chip-seq analysis for *Dt2*⁶, we have changed the motif sequences in Fig. 4b and Supplementary Fig. 14a.

The quality of the EMSA is also not very high (or the motif does not contain a good CARG-box).

Response: Thank you. We have repeated the EMSA again, and the new results should be better now (Fig. 4c and Supplementary Fig. 14b).

- It is a pity that there is only an overexpression line for *Agl22*, as MADS-domain TFs can all bind to CARG-boxes and the phenotype obtained in an overexpression line does not prove that the specific MADS-box gene actually functions in the regulation of branching number. However, this is likely based on the interaction of *Agl22* with SOC1a/*Dt2* and its presence in the axillary bud.

Response: Thank you for pointing this out. Yes, we agree with you that only the phenotype resulting from overexpression of *Agl22* is not solid enough to determine whether it is involved in branching development in soybean. We thought that because *Agl22* have multiple homologs in the soybean genome, knockout *Agl22* may not result in phenotypic changes. Therefore, we only overexpressed of *Agl22*. However, as you

pointed out, we have the interaction of *Agl22* with *SOC1a/Dt2* and the in-situ result of *Agl22* presence in the axillary bud. These results may support that *Agl22* is involved in branching development in soybean.

In the revised manuscript, we obtained another overexpression line that showed a similar result of decreased branch number as the previous overexpression line (Fig. 3i, j and Supplementary Fig. 9f), confirming that overexpression of *Agl22* indeed decreased branch number in soybean. Thank you.

- The link to Arabidopsis research concerning the role of *FUL*-like genes in branching has not been described, nor the role of *VEG1/MtFUL1-c* in other legumes (in the Introduction or Discussion section).

Response: Thank you for the suggestions. We have added the research progress of the MADS gene family, including *FUL*-like genes in *Arabidopsis*, *VEG1/PsFUC1-c* in pea and *MtFULc* in *Medicago* in the discussion section: “MADS-box transcription factor genes have been extensively studied in plants and were reported to play important roles in plant development, especially in vegetative-reproductive transition and plant architecture. For instance, *SUPPRESSOR OF OVEREXPRESSION OF CO1* and *FRUITFULL* were found to not only promote flowering, but also affect determinacy of meristems in *Arabidopsis*³. Previous studies in soybean have illuminated that the members involved in the working model of *Dt2* from this study, including *Dt2*, *GmAgl22*, *GmSoc1* and *GmAp1*, all regulate flowering time by involving in the inflorescence transition⁴⁻⁸. Similarly, *MtFUL1-c* and *VEG1/PsFUL1-c*, ortholog of *Dt2* from *Medicago* and pea respectively, were also found to play important role in the reproductive phase transition^{9,10}. Here, we found that these genes not only affect flowering time, but also control branch number in soybean. Consistent, the *Arabidopsis* homologs of *Dt2-GmSoc1a*, *FUL-SOC1*, were also found to regulate branch number^{3,11,12}. These results suggested that *Dt2* and other MADS box genes are highly pleiotropic in regulating vegetative-reproductive transition and plant architecture. In addition, these results also indicated that the network of MADS box genes in regulating plant architecture may be functional conserved among plant species. Our results

together previous studies ⁴⁻⁸ revealed that Dt2, GmAgl22 and GmSoc1 function as a complex to regulate *GmApl1*. Interestingly, the genes involved in the working model of *Dt2* all belong to the MADS-box gene family, indicating a complicate regulatory network among different MADS-box transcription factor genes although they belong to the same transcription factor family. Further investigation of more MADS-box members is needed to disclose more detailed regulatory networks of this important transcription factor gene family.” (Lines 278-300).

Thank you.

Reviewer #3 (Remarks to the Author):

The authors presented a mature and very well-written manuscript showcasing how natural variation at *Dt2* determines shoot branching and yield in soybeans. Using genetic, molecular, biochemical, and transgenic approaches authors convincingly show that Dt2 worked by interacting with GmAgl22 and GmSoc1a to physically bind to the promoter of GmApl1a and GmApl1d and to activate their transcription. Population genetic investigations showed that selection at *Dt2* was associated with geographic differentiation.

This work is highly original and reflects how few we actually know about plant architectural mechanisms in one of our major legume crops. Hence, I enjoyed reading this manuscript. I don't have considerable objections for this study. However, I still have one point for consideration and improvement:

1) I found the model authors derived in figure 5b quite appealing; however, it is not very clear which color the reproductive structures have. As far as I could comprehend AM turned to IMs are indicated by pinkish triangles. If so, then taller, more indeterminate plants have less reproductive structures than smaller ones, which is in contrast to the actual data. Please clarify the figure in a way that it becomes consistent with your findings.

Response: Thank you so much for your encouragement. We apologize for the unclear descriptions. Yes, the pinkish triangles indicate the transitions from AM to IM, we added the description in the figure legends. We originally wanted to show the final

status of inflorescence and branch in the two haplotypes, but it do have the problem as you pointed out that the previous model may lead to a confusion that more indeterminate plants have fewer reproductive structures than smaller ones. To eliminate confusion and misunderstanding, we deleted some of the pinkish triangles. Thank you so much for this important comment.

Reviewer #4 (Remarks to the Author):

The objective of Liang et al. was to identify key genes controlling branch number in soybean, which is an important shoot architecture trait related to seed yield. The authors performed a GWAS on a panel of 2573 accessions that had been re-sequenced, providing over 4M SNPs. The only association found between branch number of SNPs was in the region of *Dt2*. The authors performed gene expression analyses and found that gene expression at *Dt2* was negatively correlated with branch number. They went further creating gene edited knockouts and transgenic overexpressed lines at *Dt2* and interacting genes, showing that *Dt2* and genes producing downstream products affect branch number in soybean. The results from this study show that *Dt2* has effects beyond determinacy, affecting shoot architecture in addition to timing of vegetative to reproductive phases.

Overall, I think this is a terrific study that used many tools and thoroughly showed the important effect of *Dt2*. Quantitative genetics combined with functional analyses through genetic modification and gene expression analyses were deployed.

Response: Thank you so much for your encouragement.

Lines 28-43: Overall, the tone of this paragraph makes it sound like these advances were on the basis of design using knowledge about genotype, or basically that phenotype followed selected on genotype, when in fact many of these advances were the product of unconscious selection. In other words, changes in the genotype followed selections on the phenotype.

Response: We apologize that this paragraph makes it sound like these advances were on the basis of design using knowledge about genotype. In fact, we tried to emphasize

the important role of some key genes (or alleles) in breeding, either by unconscious or conscious selection. We also described this in the text: “By **conscious or unconscious** selection of the superior alleles of key genes that confer branching architecture, humans have significantly increased crop yields”. To indicate the correct tone, we changed the sentence to: “By **unconscious or conscious** selection of the superior alleles of key genes that confer branching architecture, humans have significantly increased crop yields”. Thank you.

Line 41: Need reference backing up statement that new superior cultivars were developed by introducing *ipa1*.

Response: Thank you so much for your suggestion. As you suggested, we listed the reference in the text: “By **introducing beneficial *ipa1* alleles into widely cultivated cultivars, a series of new elite varieties with higher yields were developed²¹**”. In this reference, the authors developed Jiayou Zhongke series of varieties using of *IPAI*^{-1D} and *IPAI*^{-2D} alleles. The developed varieties showed markedly increased yields.

Lines 72-73: Here it is claimed there is evidence that gene 18g_242900 showed higher specific expression among 16 possible candidate genes in the region identified by GWAS, but in Fig S2, this is not apparent as this gene does not seem to have a expressions profile unique at all relative to the other. Also, Fig 1 is cited here as supported evidence in expression but this data is not provided in Figure 1.

Response: Thank you for this comment. Fig 1d cited here intended to refer to the text “Within this 120-Kb interval, a total of 16 protein-coding genes were annotated according to the reference genome ZH13”. We also realized that Supplementary Fig. 2b may not be closely referring to the results here, we also removed it from the revised text. We apologize that we did not clarify this issue. According to the new results, this part was updated as follows: “**GWAS performed using a mixed linear model revealed a stable association signal across the two years in a 40-kb interval block on chromosome 18 (Fig. 1a-d and Supplementary Fig. 1d-g). Within this 40-kb interval, a total of 5 protein-coding genes were annotated according to the reference genome ZH13¹⁶ (Fig. 1d), among which *SoyZH13_18g242900* showed higher specific expression at the shoot**

apical meristem (**Supplementary Fig. 2a**), a tissue closely related to the final branching architecture¹.” (Lines 67-73).

Line 82: Why deploy a MAF cutoff of 0.1 here and not 0.05 like described in the Methods section? In general, with over 2500 accessions, a much lower MAF than 0.05 can be used in general. I think this cutoffs should be justified or altered as they can affect results and are quite arbitrary.

Response: Thank you so much for this important correction. We sincerely apologize for the typo mistake here. We deploy an MAF cutoff of 5% and a missing rate 10% to filter the SNPs as described in the methods. We have corrected it to “Of the association polymorphisms with allele frequency (MAF) greater than **0.05**, two SNPs from the promoter regions (3,259-bp and 2,580-bp upstream of the translation start site, respectively) and five SNPs from the introns showed higher association values than the threshold”. Thank you so much again.

Line 242: Here and elsewhere I think there needs to be more acknowledgment of the effect of maturity on yield with changed in *Dt2*. Many so-called yield genes just affect maturity, and it is easy to breed for increased yield by increasing maturity, but that is not practical as often a limited growing season is reality.

Response: Thank you for your suggestion. Yes, the modulation of *Dt2* could enhance soybean adaptation and yield is related to its effect on maturity. Following your suggestion, we revised the text as follows: “the *Dt2*^{CR} lines showed significantly higher yields than DN50, either at a lower planting density or a higher planting density, which was also associated with its effect on maturity (Supplementary Fig. 16b, c).” and deleted the sentence “Modulation of *Dt2* enhanced soybean adaptation and yield” from the abstract in the revised manuscript. Accordingly, we also changed the title “Natural variation of *Dt2* determines branching and enhances yield in soybean” to “Natural variation of *Dt2* determines branching in soybean”.

References:

1. Li, Y. *et al.* Genetic structure and diversity of cultivated soybean (*Glycine max* (L.) Merr.) landraces in China. *Theor. Appl. Genet.* **117**, 857-871 (2008).
2. Liu, Y. *et al.* Pan-genome of wild and cultivated soybeans. *Cell* **182**, 162-176 (2020).
3. Melzer, S. *et al.* Flowering-time genes modulate meristem determinacy and growth form in *Arabidopsis thaliana*. *Nat. Genet.* **40**, 1489-1492 (2008).
4. Kou, K. *et al.* A functionally divergent SOC1 homolog improves soybean yield and latitudinal adaptation. *Curr. Biol.* **32**, 1728-1742 (2022).
5. Yue, L., Li, X., Fang, C. & Kong, F. FT5a interferes with the Dt1-AP1 feedback loop to control flowering time and shoot determinacy in soybean. *J. Integr. Plant Biol.* **63**, 1004-1020 (2021).
6. Zhang, D. *et al.* A post-domestication mutation, *Dt2*, triggers systemic modification of divergent and convergent pathways modulating multiple agronomic traits in soybean. *Mol. Plant* **12**, 1366-1382 (2019).
7. Liu, Y. *et al.* Innovation of a regulatory mechanism modulating semi-determinate stem growth through artificial selection in soybean. *PLoS Genet.* **12**, e1005818 (2016).
8. Ping, J. *et al.* *Dt2* is a gain-of-function MADS-domain factor gene that specifies semideterminacy in soybean. *Plant Cell* **26**, 2831-2842 (2014).
9. Cheng, X., Li, G., Tang, Y. & Wen, J. Dissection of genetic regulation of compound inflorescence development in *Medicago truncatula*. *Development* **145**, dev158766 (2018).
10. Benlloch, R. *et al.* Genetic control of inflorescence architecture in legumes. *Front. Plant Sci.* **6**, doi: 10.3389/fpls.2015.00543 (2015).
11. Karami, O. *et al.* A suppressor of axillary meristem maturation promotes longevity in flowering plants. *Nat. Plants* **6**, 368-376 (2020).
12. Bemer, M. *et al.* FRUITFULL controls *SAUR10* expression and regulates *Arabidopsis* growth and architecture. *J. Exp. Bot.* **68**, 3391-3403 (2017).
13. Fang, C. *et al.* Genome-wide association studies dissect the genetic networks underlying agronomical traits in soybean. *Genome Biol.* **18**, 161 (2017).
14. Zhou, Z. *et al.* Resequencing 302 wild and cultivated accessions identifies genes related to domestication and improvement in soybean. *Nat. Biotechnol.* **33**, 408-414 (2015).

15. Shen, Y. *et al.* De novo assembly of a Chinese soybean genome. *Sci. China Life Sci.* **61**, 871-884 (2018).
16. Churchill, G.A. & Doerge, R.W. Empirical threshold values for quantitative trait mapping. *Genetics* **138**, 963-71 (1994).
17. Wei, X. *et al.* Genetic discovery for oil production and quality in sesame. *Nat. Commun.* **6**, 8609 (2015).
18. Li, Y., Du, Y., Huai, J., Jing, Y. & Lin, R. The RNA helicase UAP56 and the E3 ubiquitin ligase COP1 coordinately regulate alternative splicing to repress photomorphogenesis in *Arabidopsis*. *Plant Cell* (2022).
19. Song, X. *et al.* IPA1 functions as a downstream transcription factor repressed by D53 in strigolactone signaling in rice. *Cell Res.* **27**, 1128-1141 (2017).
20. Lykke-Andersen, S. & Jensen, T.H. Nonsense-mediated mRNA decay: an intricate machinery that shapes transcriptomes. *Nat. Rev. Mol. Cell Bio.* **16**, 665-677 (2015).
21. Zhang, L. *et al.* A natural tandem array alleviates epigenetic repression of *IPA1* and leads to superior yielding rice. *Nat. Commun.* **8**, 14789 (2017).

List of the updated figures and tables

Current version	Last version
Supplementary Fig. 1	Supplementary Fig. 1
Supplementary Fig. 2	Supplementary Fig. 2
Supplementary Fig. 3	Supplementary Fig. 3
Supplementary Fig. 4	Supplementary Fig. 4
Supplementary Fig. 5	Supplementary Fig. 5
Supplementary Fig. 6	Supplementary Fig. 6
Supplementary Fig. 7	Newly added
Supplementary Fig. 8	Supplementary Fig. 7
Supplementary Fig. 9	Supplementary Fig. 8
Supplementary Fig. 10	Supplementary Fig. 9
Supplementary Fig. 11	Supplementary Fig. 10
Supplementary Fig. 12	Supplementary Fig. 11
Supplementary Fig. 13	Supplementary Fig. 12
Supplementary Fig. 14	Supplementary Fig. 13
Supplementary Fig. 15	Newly added
Supplementary Fig. 16	Supplementary Fig. 14
Supplementary Table 1	Supplementary Table 1
Supplementary Table 1	Supplementary Table 1
Supplementary Table 3	Newly added

Reviewers' Comments:

Reviewer #1:

Remarks to the Author:

The new version of the manuscript has mostly addressed all my comments, and the overall conclusions are well supported by the numerous analyses. Nevertheless, I think that authors should review the following easy point.

- In the new version, authors still use the term "selection" as synonym of "genetic differentiation", in abstract, results and Figure legends. In my opinion, this is incorrect because genetic differentiation might be explained by selection but also by demographical history of soybean. For this reason I suggest correcting the inappropriate use of selection in the following sections:

- In Abstract, the sentence "Population genetic investigation showed that the selection of Dt2 was associated with geographic differentiation" might be rephrased to "Population genetic investigation showed that the genetic differentiation of Dt2 displayed significant geographic structure."

- Similarly, in Results section, lines 239-241, the sentence "FST analysis showed that the Dt2 locus exhibited a selection tendency between ecoregions II/I, but not between ecoregions II/III (Supplementary Fig. 15b), indicating that the differential distribution of..." could be rephrased as "FST analysis showed that the Dt2 locus exhibited a genetic differentiation tendency between ecoregions II/I, but not between ecoregions II/III (Supplementary Fig. 15b), indicating that the distinct geographic distribution of ..."

- In line 245, I suggest rephrasing the sentence "The selection of Dt2 in geographical differentiation inspired us..." to something like "The geographic and genetic differentiations of Dt2 inspired us ..."

- In the legend of Supplemental Figure 15, the title reading "Selection Detection of Dt2..." might be rephrased to "Genetic differentiation of Dt2...".

On the other hand, I suggest reviewing English language in some sections, specially the abstract, and correcting the following typographical errors:

- In Figure 4, in panel 4o, the X axis misses the name of the mutant genotype called "GmAp14m"; this should be added.

- Line 131, where it reads "An in situ hybrid assay..." should read "An in situ hybridization assay...".

- In line 211, where it reads "together showed stronger active activity..." should better read "together showed stronger activity...".

Reviewer #2:

Remarks to the Author:

In the revised version of the manuscript 'Natural variation of Dt2 determines branching in soybean' from Liang et al., the authors addressed the comments of the different reviewers. To meet my request for a better phenotypic description, the authors generated Supplementary Figure 7, which shows the "Growth state statistic of axillary buds in leaf axil between DN50 and Dt2CR lines" and write in the text "Growth state statistics of axillary buds in leaf axils between DN50 and Dt2CR lines showed that the effect of Dt2 on branch development may be related to inflorescence determinacy". While this still remains quite vague, I can deduct from the Figure that the number of "branches" (actually secondary inflorescences) is increased compared to the WT, indeed probably as a result of a delay in inflorescence shoot (primary inflorescence) termination, reminiscent of the phenotype in the *Arabidopsis ful* mutant (Balanza et al., 2018). In this new version, the authors added a reference to this paper to the Discussion section, and better embed the data in the already published legume and *Arabidopsis* literature, which is nice.

However, although I do understand that the authors identified this locus via GWAS and were before unaware that Dt2 could also regulate branch (secondary inflorescence) number, the study of Ping et al. (2014; <https://doi.org/10.1105/tpc.114.126938>) already describes that the semideterminant inflorescence phenotype of the dt2 mutant is caused by overexpression of the FULc/VEG1 homolog Dt2 in the inflorescence apex, in line with the phenotype in 35S:CDS-Dt2 lines, which have less

"nodes" (secondary inflorescences) and a lower plant height. Thus, although Liang et al. write that the increased branch number has not been described before, this is not the case as it has already been described by Ping et al., albeit named differently as a 'semideterminate inflorescence shoot'. It is interesting that in both the Ping et al. and Liang et al. studies, the prolonged production of secondary inflorescences at lower Dt2 levels seems to be accompanied by longer internode length and vice versa. In conclusion, my opinion that the described phenotype is not new for Dt2 has been strengthened and although the authors can couple it here to natural variance in the Dt2HapI/HapII haplotypes, the study has not lead to many new insights in the function of Dt2. I still think that it is a solid study however, which nicely confirms using GWAS that Dt2 is a pleiotropic gene that is important for both flowering time, inflorescence structure and inflorescence shoot determinacy (and thereby branch number). The impact of the current study could be increased by further investigation of the cause of the differences between the HapI and HapII, but this will indeed be a lot of work.

Reviewer #3:

Remarks to the Author:

Thank you for considering my points.

REVIEWERS' COMMENTS

Reviewer #1 (Remarks to the Author):

The new version of the manuscript has mostly addressed all my comments, and the overall conclusions are well supported by the numerous analyses. Nevertheless, I think that authors should review the following easy point.

Response: Thank you so much for the encouragement. We appreciate your previous and this time constructive comments. We have revised the manuscript as you suggested, and we hope it will address all your questions. Thank you again.

- In the new version, authors still use the term “selection” as synonym of “genetic differentiation”, in abstract, results and Figure legends. In my opinion, this is incorrect because genetic differentiation might be explained by selection but also by demographical history of soybean. For this reason I suggest correcting the inappropriate use of selection in the following sections:

-In Abstract, the sentence “Population genetic investigation showed that the selection of *Dt2* was associated with geographic differentiation” might be rephrased to “Population genetic investigation showed that the genetic differentiation of *Dt2* displayed significant geographic structure.”

Response: It has been revised as you suggested. Thank you.

- Similarly, in Results section, lines 239-241, the sentence “FST analysis showed that the *Dt2* locus exhibited a selection tendency between ecoregions II/I, but not between ecoregions II/III (Supplementary Fig. 15b), indicating that the differential distribution of...” could be rephrased as “FST analysis showed that the *Dt2* locus exhibited a genetic differentiation tendency between ecoregions II/I, but not between ecoregions II/III (Supplementary Fig. 15b), indicating that the distinct geographic distribution of ...”

Response: It has been revised as you suggested. Thank you.

- In line 245, I suggest rephrasing the sentence “The selection of *Dt2* in geographical differentiation inspired us...” to something like “The geographic and genetic differentiations of *Dt2* inspired us ...”

Response: It has been revised as you suggested. Thank you.

- In the legend of Supplemental Figure 15, the title reading “Selection Detection of Dt2...” might be rephrased to “Genetic differentiation of Dt2...”.

Response: It has been revised as you suggested. Thank you.

On the other hand, I suggest reviewing English language in some sections, specially the abstract, and correcting the following typographical errors:

- In Figure 4, in panel 4o, the X axis misses the name of the mutant genotype called “GmAp14m”; this should be added.

Response: Appreciate you so much for the careful review. It has been added as you suggested. Thank you.

- Line 131, where it reads “An in situ hybrid assay...” should read “An in situ hybridization assay...”.

Response: It has been revised as you suggested. Thank you.

- In line 211, where it reads “together showed stronger active activity...” should better read “together showed stronger activity...”.

Response: It has been revised as you suggested.

Sincerely appreciate your constructive comments again.

Reviewer #2 (Remarks to the Author):

In the revised version of the manuscript ‘Natural variation of Dt2 determines branching in soybean’ from Liang et al., the authors addressed the comments of the different reviewers. To meet my request for a better phenotypic description, the authors generated Supplementary Figure 7, which shows the “Growth state statistic of axillary buds in leaf axil between DN50 and Dt2CR lines” and write in the text “Growth state statistics of axillary buds in leaf axils between DN50 and Dt2CR lines showed that the effect of Dt2 on branch development may be related to inflorescence determinacy”. While this still remains quite vague, I can deduct from the Figure that the number of “branches” (actually secondary inflorescences) is increased compared to the WT, indeed probably

as a result of a delay in inflorescence shoot (primary inflorescence) termination, reminiscent of the phenotype in the *Arabidopsis ful* mutant (Balanza et al., 2018). In this new version, the authors added a reference to this paper to the Discussion section, and better embed the data in the already published legume and *Arabidopsis* literature, which is nice.

However, although I do understand that the authors identified this locus via GWAS and were before unaware that Dt2 could also regulate branch (secondary inflorescence) number, the study of Ping et al. (2014; <https://doi.org/10.1105/tpc.114.126938>) already describes that the semideterminant inflorescence phenotype of the dt2 mutant is caused by overexpression of the FULc/VEG1 homolog Dt2 in the inflorescence apex, in line with the phenotype in 35S:CDS-Dt2 lines, which have less “nodes” (secondary inflorescences) and a lower plant height. Thus, although Liang et al. write that the increased branch number has not been described before, this is not the case as it has already been described by Ping et al., albeit named differently as a ‘semideterminate inflorescence shoot’. It is interesting that in both the Ping et al. and Liang et al. studies, the prolonged production of secondary inflorescences at lower Dt2 levels seems to be accompanied by longer internode length and vice versa.

In conclusion, my opinion that the described phenotype is not new for Dt2 has been strengthened and although the authors can couple it here to natural variance in the Dt2HapI/HapII haplotypes, the study has not lead to many new insights in the function of Dt2. I still think that it is a solid study however, which nicely confirms using GWAS that Dt2 is a pleiotropic gene that is important for both flowering time, inflorescence structure and inflorescence shoot determinacy (and thereby branch number). The impact of the current study could be increased by further investigation of the cause of the differences between the HapI and HapII, but this will indeed be a lot of work.

Response: Thanks for your professional insight. As we all known, during the vegetative phase, the shoot apical meristem (SAM) generates leaf primordia and further axillary vegetative shoots that initiated from the axillary meristem (AM)¹. By sensing the flowering induction signals in the surrounding environment signal, plant change the differentiation status of some cells in the SAM or lateral meristem (LM), into inflorescence meristem (IM) which further develops into floral meristem (FM), and finally complete the transition process from vegetative growth to reproductive growth².

Lateral meristems generate branches and inflorescence structures, which define the overall form of a plant³. Thus, AM and IM are two related but distinct different tissues. The study of Ping et al. (2014; <https://doi.org/10.1105/tpc.114.126938>) had shown an influence of *Dt2* on inflorescence meristem development. In this study, we focused more on the influence of *Dt2* on axillary meristem development. Therefore, our result is a good complementary to the previous study, which further elucidate the function of *Dt2*. Yes, we agree with you that the future further investigation of the cause of the differences between the HapI and HapII, either from us or from other laboratories, will make the function of *Dt2* clearer.

Reviewer #3 (Remarks to the Author):

Thank you for considering my points.

Response: Thank you so much for the encouragement. We appreciate your constructive comments.

1. Benlloch, R. *et al.* Genetic control of inflorescence architecture in legumes. *Front. Plant Sci.* **6**, doi: 10.3389/fpls.2015.00543 (2015).
2. Luo, Y., Guo, Z. & Li, L. Evolutionary conservation of microRNA regulatory programs in plant flower development. *Developmental Biology* **380**, 133-144 (2013).
3. Gallavotti, A. *et al.* The role of barren stalk1 in the architecture of maize. *Nature* **432**, 630-635 (2004).